# Enhanced photocatalysis and biomolecular sensing with field-activated nanotube-nanoparticle templates

Sawsan Almohammed [1,2], Sebastian Tade Barwich[3], Andrew K. Mitchell [1], Brian J. Rodriguez [1,2] & James H. Rice[1]

The development of new catalysts for oxidation reactions is of central importance for many industrial processes. Plasmonic catalysis involves photoexcitation of templates/chips to drive and enhance oxidation of target molecules. Raman-based sensing of target molecules can also be enhanced by these templates. This provides motivation for the rational design, characterization, and experimental demonstration of effective template nanostructures. In this paper, we report on a template comprising silver nanoparticles on aligned peptide nanotubes, contacted with a microfabricated chip in a dry environment. Efficient plasmonic catalysis for oxidation of molecules such as p-aminothiophenol results from facile trans-template charge transfer, activated and controlled by application of an electric field. Raman detection of biomolecules such as glucose and nucleobases are also dramatically enhanced by the template. A reduced quantum mechanical model is formulated, comprising a minimum description of key components. Calculated nanotube-metal-molecule charge transfer is used to understand the catalytic mechanism and shows this system is well-optimized.

[1] School of Physics, University College Dublin, Belfield, Dublin 4 D04 V1W8, Ireland. [2] Conway Institute of Biomolecular and Biomedical Research, University College Dublin, Belfield, Dublin 4 D04 V1W8, Ireland. [3] School of Physics, Trinity College Dublin, Dublin 2 D02 PN40, Ireland. Correspondence and requests for materials should be addressed to A.K.M. (email: andrew.mitchell@ucd.ie) or to B.J.R. (email: brian.rodriguez@ucd.ie) or to J.H.R. (email: james.rice@ucd.ie)

Many industrial processes require efficient catalysis of redox reactions[1–4]. A seemingly different problem is the sensing of biomolecules, which can pose its own challenges[4–6]. In fact, these scientific problems are related—both catalytic activity and sensing can be dramatically enhanced by the use of nanostructured templates/chips. In the field of plasmon catalysis, a major goal of current research is to design such templates to facilitate charge transfer to target molecules, thereby enhancing plasmonic photocatalytic activity[1–4] and enhancing the Raman signal intensity of otherwise spectroscopically dark molecules[4–6].

Nanomaterials that exhibit elevated catalytic performance play a significant role in high value industrial processes such as the reduction of carbon dioxide or nitroaromatics[1–4]. Catalytic processes augmented by plasmon active nanomaterials for example gold (Au) or silver (Ag) offer significant potential to enhance heterogeneous catalytic processes[4–6]. The catalytic enhancement associated with metal nanoparticles (NPs) results from optical excitation of localized surface plasmon resonances (LSPRs) that act as the energy input to drive chemical transformations in redox reactions, and may also be facilitated by the intense local electric field and local heat generated[1–4]. The specific mechanisms of LSPR-mediated photocatalytic processes are not fully understood, but are investigated in detail in this paper. It is widely believed that hot electron processes play a significant role in the transferal of chemical, photon, and electrical energies during plasmon-mediated catalysis[2–4]. The correlation between hot electron flow and the rates of photocatalytic reactions has been reported in several studies revealing that hot electrons strongly influence heterogeneous photocatalysis[2–6]. Through setting the wavelength of incident radiation to match the HOMO–LUMO gap of attached molecules, charge transfer to the molecule can occur, giving rise to oxidation reactions in certain molecules or enhanced Raman scattering cross section for sensing in other molecules.

Nanocomposites combining semiconductor materials with metal NPs bring additional benefits to plasmonic photocatalysis[6–9]. The contact potential difference between metal NPs and a semiconductor can separate photogenerated electrons and holes[6–9], thereby reducing electron-hole pair diffusion lengths and leading to more efficient photogenerated charge separation and transfer, which in turn enhances photocatalytic activity[2–5].

Here, we show an increase in probability and efficiency of both chemical reactions and SERS detection through electro-optical synergy, using a microfabricated chip design (in air rather than electrochemical)[2–4]. This is achieved through the use of a plasmonic-semiconductor system based on aligned diphenylalanine peptide nanotube (FF-PNTs) wide band gap semiconductors[10–15]. Diphenylalanine (FF), a peptide consisting of a naturally occurring amino acid phenylalanine, can self-assemble into micro and nanosized tubular structures[10–17]. The resulting organic self-assembled nanotubes are the FF-PNTs, which are stable/robust, rigid, and also biocompatible. They can be used in applications requiring the use of a wide bandgap semiconductor[10–17]. FF-PNTs have been reported to have high thermal and chemical stability[10–15] in addition to piezoelectric[16,17] and pyroelectric[15] properties. We show experimentally and theoretically that applying a longitudinal electric field allows the FF-PNT density of states to be tuned from a semiconductor to a metal, enabling effective charge transfer from the nanotube to the metal nanoparticles. This results in an enhancement in the state density of hot electrons[18–20]. The effect is optimized through the physical alignment of the FF-PNTs, since the inherent electric dipoles of the FF-PNTs are then aligned and maximally responsive to the applied longitudinal electric field (a cooperative effect). We demonstrate that this optoelectrical device enhances photocatalytic conversion for model oxidation reactions

(exemplified here by p-aminothiophenol (PATP) oxidized to p-nitrothiophenol (PNTP), and 2-aminophenol (2-AMP) oxidized to 2-nitrophenol (2-NIP)), exploiting the facile field-activated trans-template charge transfer. We also demonstrate that this same approach can be used to enhance the strength of Raman scattering from molecules with small Raman cross-sections for example glucose and DNA-based molecules, establishing the potential of our template design for sensitive detection and analytics. This approach is versatile and can be applied to a range of plasmonic metal nanoparticle and semiconductor combinations.

## Results

**Photocatalysis studies.** The template comprises microfabricated gold electrodes on an Si substrate, with a 0.1 mm opening size between electrodes. FF-PNTs are aligned during self-assembly on an optically-thick insulating layer of $SiO_2$ 1 mm wide, exploiting Si/SiO₂ wettability differences[21,22], and Ag NPs deposited (Fig. 1a, Supplementary Fig. 1). Experiments are undertaken in a dry environment (not electrochemical). Scanning electron microscopy (SEM) images (Fig. 1b–d and Supplementary Fig. 2) show the alignment of the FF-PNT and the topology/morphology of deposited Ag NPs, which aggregate around the FF-PNTs due to functional-group interactions[21,22]. An external electric field is produced by applying a voltage across the electrodes. The current measured through the FF-PNT/Ag NPs template due to the applied voltage reveals Ohmic behavior (Supplementary Fig. 3), with a resistance ($658 \pm 0.77\ \Omega$) lower than for Ag NPs ($1583 \pm 0.92\ \Omega$) or for FF-PNTs ($930 \pm 1.03\ \Omega$) alone, consistent with previous studies[23]. These results further demonstrate the feasibility of using FF-PNTs in electronic devices, and provide a platform to explore the influence on SERS of an applied electric field.

We first investigate the impact of applying an electric field on the oxidation reaction PATP → PNTP using the FF-PNT/Ag NP template, as monitored by SERS using an excitation wavelength fixed at 532 nm. The Raman and SERS spectra for PATP (Supplementary Fig. 4) is well characterized and is known to undergo photochemical changes in the presence of plasmonic metals[24–27]. The electric field is generated by applying a voltage across the gold nanoelectrodes (see Methods). For an offset bias of <~100 mV, a seven-fold increase in SERS signal intensity was observed, but with no change in the relative positions of the SERS spectral peaks (Supplementary Fig. 5). However, with an offset bias >~100 mV, the eight-fold increase in SERS signal strength was accompanied with differences in the vibrational mode

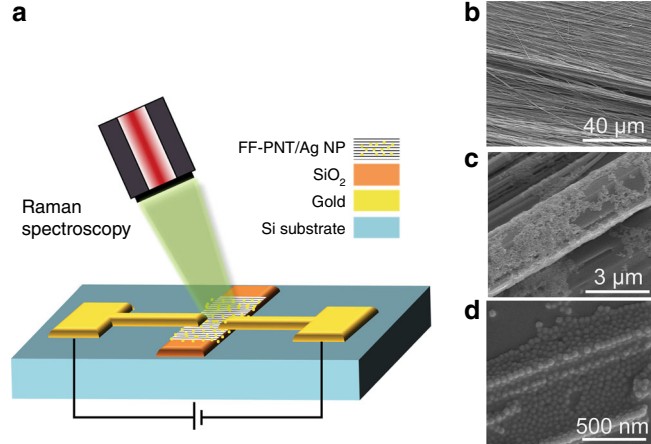

**Fig. 1** Substrate fabrication and characterization. **a** Schematic sketch of the template design that is used in the study. **b–d** Scanning electron microscopy images of the aligned FF-PNT/Ag NP structures located between the gold electrodes

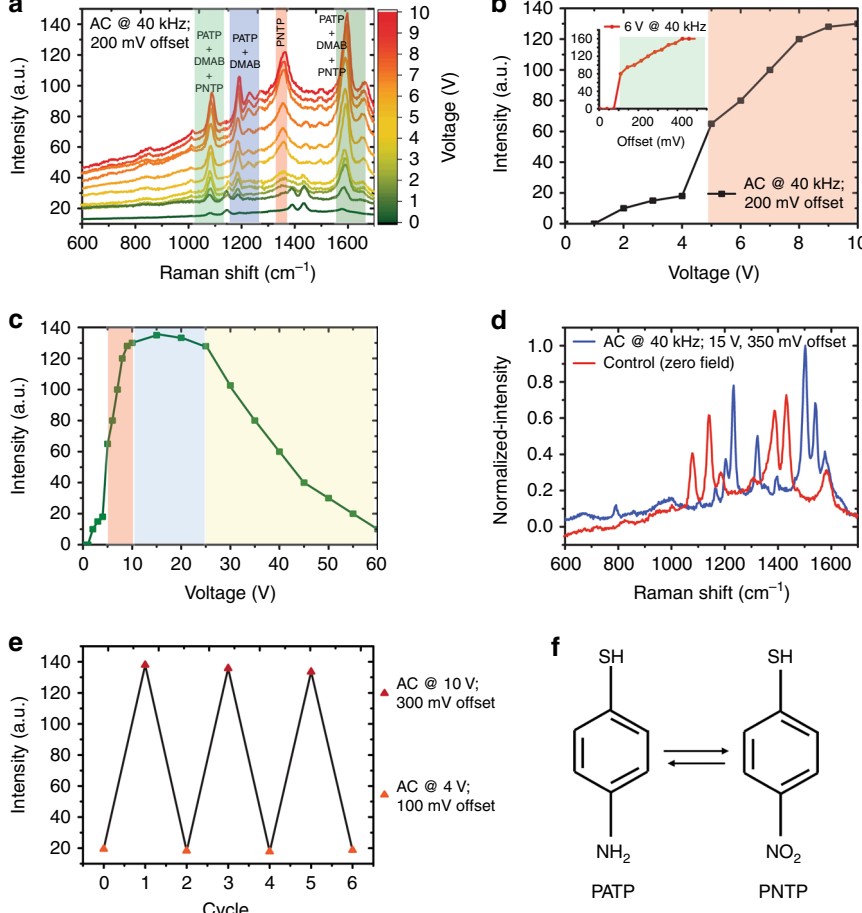

**Fig. 2** Influence of applied electric field on the SERS spectra of PATP on the template. **a** SERS measurements of PATP on the FF-PNT/Ag NP template at different electric fields strengths, applying voltages from 0–10 V at 40 kHz AC with offset DC bias of 200 mV. **b**, **c** Corresponding Raman band intensity at 1334 cm$^{-1}$ (assigned to PNTP) as a function of voltage. **b** Critical threshold voltage ~4 V required for the formation of PNTP. Inset illustrates SERS intensity with offset DC bias. **c** The window of 10–20 V is optimal for PNTP product yield via photocatalysis, with SERS intensity reducing above ~25 V. **d** Normalized SERS spectra, comparing a fully relaxed control sample at zero field (red) to the spectrum recorded with applied field (blue). **e** Plot of SERS signal strength intensity vs. AC electric field, showing manipulation of the product yield achieved by utilizing an electric field with a voltage of 10 V and an offset DC bias of 300 mV (conditions that yield PNTP) vs. a weaker electric field formed using a voltage of 4 V and an offset DC bias of 100 mV (conditions that yield PATP). **f** Schematic illustration of the transformation and switch of the product formation of PATP to PNTP

distribution (Fig. 2a). In particular, new vibrational modes at 1334 and 1376 cm$^{-1}$ were assigned to symmetric NO$_2$ stretching vibrations of the oxidation product PNTP[24–26]. Formation of PNTP can therefore be monitored using SERS through the intensity of the 1334 cm$^{-1}$ Raman band. The inset to Fig. 2b shows that the oxidation reaction is abruptly switched on at a threshold offset bias around 100 mV. We also find that the yield of the oxidized product PNTP increases with an AC voltage applied in addition to the offset bias, allowing to optimize reaction conditions. Figure 2b, c show the band intensity at 1334 cm$^{-1}$ for 200 mV offset bias as a function of AC voltage, demonstrating abrupt threshold behavior at around 4–6 V AC, while very large voltages >25 V see a decrease in band intensity (and hence product yield). Optimal conditions are found to be 100–200 mV offset bias, combined with 10–25 V AC at 40 kHz (Supplementary Fig. 5–7). Comparing the SERS spectra recorded before and following application of the electric field (Fig. 2d) shows that Raman bands associated with the starting material (PATP) are entirely replaced by Raman bands associated with the product (PNTP). Switching off the electric field results in relaxation of the SERS spectrum of PNTP back to that of PATP (Supplementary Fig. 8). The transformation of PATP to/from

PNTP can in fact be finely controlled using the applied electric field, and is fully reversible. Reactant and product can be interconverted by switching on and off the field over many cycles (with no apparent degradation), Fig. 2e. We note that applying a pure AC voltage (with zero offset bias) does not result in the oxidation of PATP to PNTP (Supplementary Fig. 7), although SERS intensity of the PATP bands is enhanced.

To additionally assess the properties of the FF-PNT/Ag NP template with respect to plasmonic catalysis, we studied the impact of broadband super bandgap irradiation ($\lambda_{ex}$ centered at 255 nm). Photoexciting the nanotubes directly increases the photocatalytic reaction processes, as well as the SERS signal intensity (Supplementary Fig. 9). The increase in SERS signal strength occurred with transformations in the spectral profile, which included the appearance of a vibrational mode at 1334 cm$^{-1}$ assigned to the occurrence of PNTP[24–26]. It is known that super bandgap excitation of a Au NPs/TiO$_2$ composite enhances the plasmonic photocatalytic oxidation of PATP to PNTP through charge transfer processes, where electrons from UV excited TiO$_2$ transfer to the Au NPs and then to PATP[24–27]. The UV-excited electrons from TiO$_2$ and the hot electrons from the Au NPs take part in the O$_2$ activation step to yield PNTP[24–27]. The formation of PNTP from PATP using our

FF-PNT/Ag NP template might be expected to result from a similar mechanism, where fermi level alteration of Ag NPs occurs via charge transfer of photo-generated electrons from FF-PNTs following super band gap irradiation. This creates conditions for a photocatalyzed oxidation reaction of PATP to PNTP[28]. We find that either an external electric field or UV irradiation treatment of the FF-PNT/Ag NP template catalyzes the oxidation reaction—and in fact that the combination of both electric field and UV irradiation further enhances product yield (Supplementary Fig. 9).

As a control study, PATP attached to Ag NPs on a simple Si substrate (in the absence of FF-PNTs) were studied. The resulting SERS spectra show no changes in the band positions with applied external electric field, although intensity fluctuations were observed (Supplementary Fig. 10). By contrast, SERS measurements on the FF-PNT/Ag NP template in the absence of the probe molecule (Supplementary Figs. 11 and 12) show a ~four-fold intensification in SERS signal intensity, together with strong spectral blinking. This indicates that charging of the FF-PNT/Ag NP complex results from applying an electric field. Previous studies suggest that an external electric field can additionally enhance the localized electric field produced by Ag NP as the electrons are disturbed from their equilibrium positions, resulting in Raman intensity fluctuations[18–20]. Another possible reason for the appearance of fluctuations in FF-PNT spectra is that the oscillating electric field may induce synchronous FF-PNT molecular vibrations for high voltage and offset bias that can change their dipole moment[16,17,29,30].

Raman control studies of a probe molecules on FF-PNTs in the absence of Ag NPs show a modest increase in signal intensity (possibly indicating partial charge redistribution), but no changes in band positions, establishing the need for the LSPRs from the Ag NPs for oxidation reactions.

We also performed extensive tests to establish the high stability of the template. SEM imaging of the sample showed spherical features assigned to Ag NPs. These Ag NPs occur in clusters predominately situated above and amid the FF-PNTs (Fig. 1, Supplementary Fig. 13). We note that following the addition of the probe molecule, these structural characteristics remain unaltered. Following application of the electric field, recorded SEM images also showed no signs of changes in geometrical topologies and morphologies of the Ag NPs or FF-PNT/Ag NP template.

Optical absorption spectroscopy measurements were performed to investigate changes in Ag NP electron density on the FF-PNT template using an applied electric field (Supplementary Fig. 15). Monitoring the LSPR absorption band with increasing electric field showed a broadening and a red shift in the LSPR band ($\Delta_{shift}$ ~20 nm). Noting that no change in the LSPR band was seen for an applied field on the Ag NPs only (Supplementary Fig. 14), this demonstrates that the FF-PNTs influence how the electric field interacts with the LSPR of the Ag NPs[17,21,22]. The broadening in the LSPR band of the Ag NPs may result from aggregation of the NPs on the tubes (Fig. 1, Supplementary Fig. 3), while the red shift can be accounted for a buildup in Ag NP electron density during irradiation that alters the NP refractive index. The modification of the Ag NPs electron density ($\Delta N/N$) was estimated to be 9%, following an applied electric field resulting in a 20 nm red shift (where $\Delta N/N = 2\Delta\lambda/\lambda_0$, $\Delta\lambda$ being the measured wavelength modification, and $\lambda_0$ is the original Ag NPs plasmon peak location)[31]. This is equivalent to the density reported for a FF-PNT/Ag NP template under UV wavelength irradiation, which was assigned to photogenerated electrons being relocated from the FF-PNTs to the Ag NPs as they reach charge equilibration[28]. These effects where reproducible for different metal nanoparticle sizes.

Finally, we explored the effect of NP size (Supplementary Fig. 16). Although smaller NP size is known to favor hot-electron production and hence catalytic activity, SERS sensitivity is enhanced by larger NP size. Therefore, in the present work we use 60 nm NPs, which produce significant hot-electron yield while retaining good SERS sensitivity.

**Mechanism for catalysis and SERS enhancement.** The plasmonic photocatalytic pathway without an external electric field may result from the formation of hot electrons following the excitation of the Ag NPs LSPR by the Raman excitation laser. These hot electrons are reallocated to adsorbed $O_2$ molecules, generating triplet diradical $^3O_2$ formed in degenerate antibonding $\pi_{2px}$ and $\pi_{2py}$ orbitals, that subsequently is involved in the oxidation of PATP to DMAB (Fig. 3)[24–27]. Studies have reported that based on the spectral similarity of PATP and DMAB, the non-$a_1$-type peaks can be ascribed to N–N stretching of DMAB generated from PATP via a photocatalytic coupling reaction[24–26]. Additional electron(s) added to the $\pi^\star$ level (LUMO) of the PATP molecule from the Ag NPs may weaken the resonance structure, reducing the bond order. This can result in downshifts in the vibrational modes that are sensitive to this resonance structure. Specifically, these are the C=C stretching 1575 and the C–S stretching 1082 cm$^{-1}$ peaks for PATP, which shift to 1580 cm$^{-1}$ and 1077 cm$^{-1}$, respectively (Fig. 2a). The frequency shifts can be understood as resulting from the strong interaction between the adsorbate molecule and substrate. Similar behavior of such interaction between PATP and a substrate, as well as the downshifts in the vibrations was previously reported when using TiO$_2$ with metal NPs[24–26]. The application of a sufficiently strong electric field to the FF-PNT/Ag NP template thus forms PNTP from

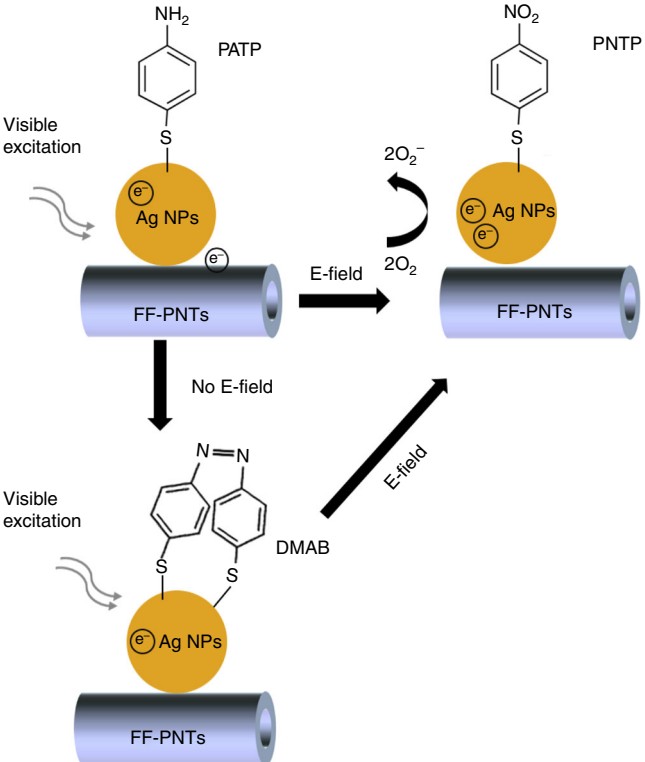

**Fig. 3** Mechanism for oxidation of PATP to PNTP on the FF-PNT/Ag NP template. When the FF-PNT/Ag NP template was excited by the Raman laser in the presence of an electric field, LSPR-excited hot electrons and additionally electrons relocated from FF-PNTs to Ag NPs contribute to the $O_2$ activation step, resulting in the creation of PNTP rather than DMAB. Also shown also is the formation of DMAB on the FF-PNT/Ag NP template, which occurs at zero field. The DMAB dimer may subsequently convert to PNTP upon application of an electric field, together with optical excitation of the Ag NPs LSPR

PATP rather than oxidizing PATP to DMAB (Fig. 3). We note that there is an alternative reaction pathway, in which a DMAP dimer converts directly to PNTP with sufficiently strong applied electric field via optical excitation of the Ag NPs LSPR (Fig. 3).

To better understand the mechanism for template-mediated photocatalysis of PATP oxidation, and for optimization of the template for future rational design, we turn now to the theory of NT-NP-molecule charge transport. Photoexcitation of the Ag NPs produces hot electrons; the wavelength of the incident radiation is tuned so the excitation energy of these hot electrons matches the HOMO–LUMO gap of the attached target molecule, thereby promoting an oxidation reaction. However, this process alone cannot be catalytically sustained without replenishing the lost electrons from the Ag NPs. The choice of substrate for the Ag NPs is therefore critical to the design of an effective template for photocatalysis. While a metallic substrate is an ideal source for NP charge replacement, the plasmonic properties of the NP would be drastically changed and any hot electrons produced would be quickly lost to the thermal conduction electron reservoir. An insulating substrate preserves the photoexcitation-oxidation mechanism, but obviously hinders template charge redistribution. In principle, however, semiconductors allow for a separation of time scales for the NP-molecule oxidation reaction, and the substrate-NP charge equilibration.

We argue that the optimal setup involves careful matching of the substrate semiconductor band-gap to the metallic NP band-width. On one hand, we aim to preserve an efficient NP-molecule plasmon-driven oxidation, while on the other hand allowing for a catalytic cycle through subsequent redistribution of charge across the semiconductor and NPs. Low-energy (non-excited) electrons of the semiconductor substrate transferred to the metallic NPs quickly relax. In general, a small finite conduction band overlap and weak coupling is required.

In this work, we use FF-PNTs as the semiconducting substrate, whose band-gap of ~4.6 eV matches approximately the Ag NP bandwidth, and so are rather well decoupled electronically at experimentally relevant temperatures. However, as was shown in Fig. 2, use of an electric field along the length of the FF-PNTs activates and dramatically enhances photocatalysis of PATP oxidation. As shown below, this is due to a field-induced coupling of low-energy electrons in the FF-PNT with the Ag NPs, opening a channel for facile charge transport across the template. The in-situ controllability of the product yield using a tunable electric field has obvious practical utility, and also opens the possibility to suppress unwanted side-reactions.

In the following, we develop a theory of charge transport across the template, to substantiate the general principles discussed above. Our philosophy here is not to undertake a realistic simulation of the full system. Even if exact first-principles calculations for such a system was possible, the significant structural and chemical complexity of the FF-PNT alone obfuscates the underlying mechanisms, largely limiting use to a case-by-case basis. Rather, we seek to develop the simplest possible quantum-mechanical model that encapsulates the vital physics of the constituents, which can then be solved exactly by quantum many-body techniques to understand in detail the key ingredients for template-enhanced plasmonic catalysis. Our model will comprise physically-motivated reduced descriptions of the FF-PNT, the Ag NPs, and the PATP molecule.

Figure 4 outlines schematically the justification for our reduced model. Figure 4a illustrates the self-assembled FF-PNT super-structure[32]: a tube-like hexagonal array of individual PNTs shown in Fig. 4a, b. FF units (Fig. 4c) are stacked to form a tube held together by CO…HN hydrogen bonds (and other non-bonded interactions)[32], as illustrated in Fig. 4d–f. The tubes can therefore

be regarded as (coupled) 1d chains of alternating strong bonds (C=O) and weak bonds (O…H) running longitudinally. Our reduced model of the FF-PNT (e) is highly simplified, consisting of a single infinite 1d quantum tight-binding chain with alternating couplings. The NT Hamiltonian in second-quantized notation reads,

$$H_{NT} = \sum_{n=-\infty}^{\infty} \left[ (t+\Delta)c_{n,a}^{\dagger}c_{n,b} + (t-\Delta)c_{n,b}^{\dagger}c_{n+1,a} + \text{H.c.} \right] + \sum_{n=-\infty}^{\infty} \sum_{\alpha} n\delta V c_{n,\alpha}^{\dagger}c_{n,\alpha},$$

(1)

where $c_{n,\alpha}^{\dagger}$ creates an electron on sublattice $\alpha = a,b$ in unit cell $n$, and $c_{n,\alpha}$ is the corresponding annihilation operator. Electron interactions[33] are neglected. The effect of an aligned static electric field is taken into account via a potential gradient in the last term of Eq. 1. The NT model is a semiconductor at zero field, as required, with a spectral gap of $4\Delta$. In the following, we set $4\Delta = 4.6$ eV to match experiment, and $t = 2.5$ eV. The gapped NT local density of states (see Methods) is shown in Fig. 5a for zero field. Figure 5b–d show spectral weight redistribution into the gap when an electric field is applied.

The model for the Ag NP is taken to be a finite 3d tight-binding lattice with typical experimental dimensions (approximately $N = 75$ Ag atoms in diameter). The NPs are large enough to be metallic, but their finite size is a critical feature that must be retained in the model to understand change transfer across it from NT to molecule. Specifically,

$$H_{NP} = t' \sum_{\langle \vec{r}, \vec{r}' \rangle} \left[ d_{\vec{r}}^{\dagger} d_{\vec{r}'} + \text{H.c.} \right],$$

(2)

where $d_{\vec{r}}^{\dagger}$ creates a NP electron at site $\vec{r} \equiv (x,y,z)$, with $\{x,y,z\} = 1,2,\ldots N$. Here $\langle \ldots \rangle$ indicates nearest neighbors, taken to be on a cubic lattice for simplicity. The NP bandwidth $12t' \simeq 4$ eV is again fit to experiment. Finally, we use a standard tight-binding model for the target molecule backbone, $H_{mol}$, with parameters chosen to reproduce the HOMO–LUMO gap. These three components are coupled in the sequence NT-NP-molecule, via local hopping matrix elements $V_{NT-NP}$ and $V_{NP-mol}$ (for further details, see Methods). Despite its simplicity, we show below that this quantum model describes the physics controlling the functionality of the template.

The central quantity of interest for our theoretical analysis is the differential conductance $G = dI/dV$ for quantum transport of electrons from the NT to the molecule. A high conductance indicates facile charge transport across the template, which is necessary for its catalytic function in oxidation reactions. We do not simulate the plasmonic physics directly per se, but focus instead on the charge redistribution following photoexcitation, encoded in the conductance. We employ the Landauer–Büttiker[34–36] formalism within linear response to calculate the conductance from NT to molecule across the NP,

$$G = \frac{2e^2}{h} \int dE \left( -\frac{df}{dE} \right) T(E),$$

(3)

where $f$ represents the Fermi function and $T(E)$ the transmission function at energy $E$. In the following we are interested in temperatures $T \ll t$. In terms of Green's functions, the transmission function can be expressed exactly as[34–36],

$$T(E) = 4\pi\Gamma V_{NT-NP}^2 A_{NT}^0 \times |\mathcal{G}_{NT-mol}|^2,$$

(4)

where $A_{NT}^0 = -\frac{1}{\pi} Im[\mathcal{G}_{NT}^0]$ is the free NT density of states at the NP as shown in Fig. 5(a–d), and $\mathcal{G}_{NT-mol}$ is the electronic propagator across the template, from NT to molecule, in the

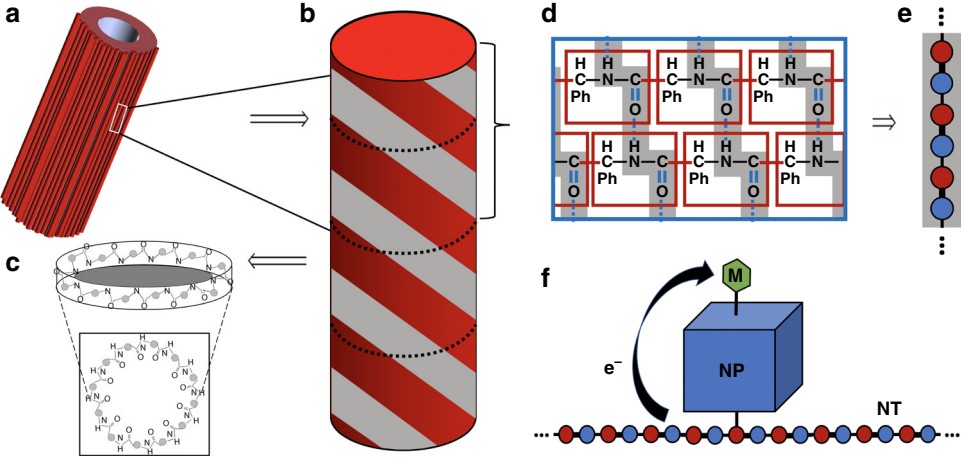

**Fig. 4** Schematic formulation of reduced theoretical model. **a** Self-assembled FF-PNT superstructure of (**b**) individual tubes, each of which is built up from (**c**) stacked six-fold macrocycles (FF comprises a ring of 6 diphenylalanine units). **d** The FF units in adjacent layers are hydrogen bonded, resulting in (**d**) the corkscrew structure and longitudinal chains with alternating strong (C=O) and weak (O...H) bonds. **e** Reduced model for the FF-PNT consisting of a single 1d chain with alternating couplings. **f** Model for the entire system, with the target molecule attached to a metallic NP, which itself is coupled to the NT chain

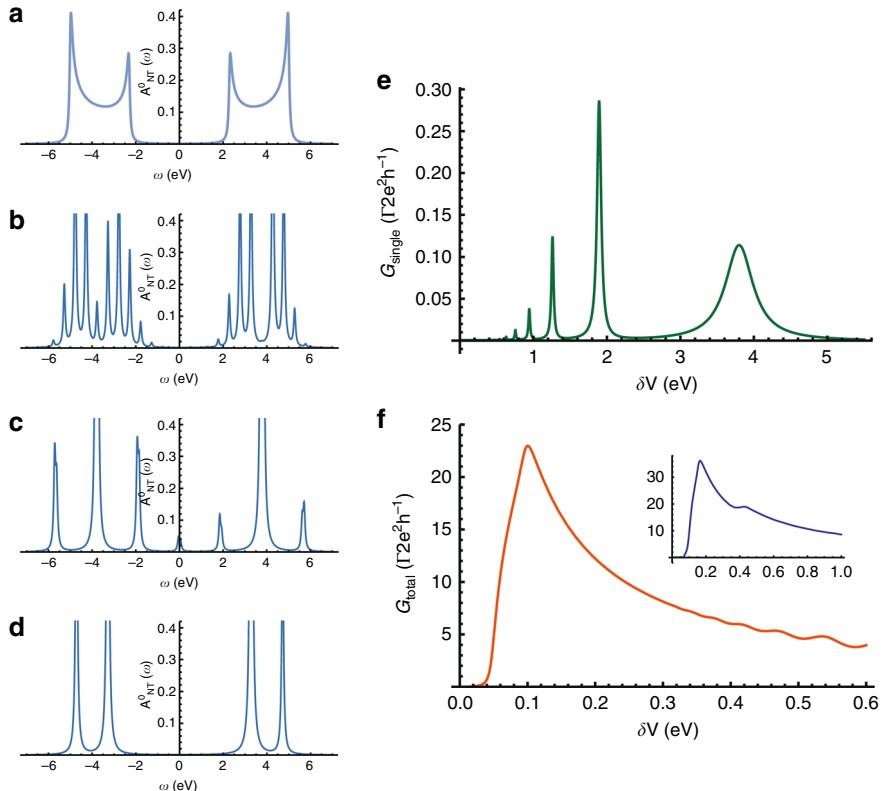

**Fig. 5** Theoretical results obtained from reduced model. **a–d** Local density of states of the NT model, Eq. 1, at the point of attachment to a NP. No electric field is applied in **a**; it is a perfect semiconductor with spectral gap 4.6 eV. Potential difference $\delta V = 0.5, 2.0, 8.0$ eV in **b**, **c**, **d**, respectively, due to an applied field. Spectral weight redistribution upon increasing field strength first yields conducting, then insulating, behavior. **e** Conductance for electron transport from the NT to the molecule across a single NP, as a function of applied field. **f** Collective conductance due to a dense array of NPs along the NT, showing threshold activation and significant enhancement of charge transport at moderate field strengths, then attenuation upon further increasing the field strength. Inset: qualitatively similar behavior for a more realistic model of the FF-PNT taking into account the six-fold macrocycle FF unit cell and inter-layer hydrogen bonding

fully coupled model. These Green's functions are calculated exactly[37,38], as described in Methods. $\Gamma \ll 1$ is the hybridization to a fictitious probe electrode which we introduce only to define a conductance (results for conductance in Fig. 5 are given in units of $\Gamma$).

The conductance across the template as a function of applied potential difference $\delta V$ (and hence longitudinal NT electric field) when a single NP is attached to the NT, is presented in Fig. 5e. It shows a series of conductance resonances as the field is tuned, indicating that there are specific values of the field strength that strongly enhance catalytic activity of the template. Closer analysis

also reveals that there is an threshold field, with $\delta V/t \sim 0.1$, below which charge transfer is blocked.

To understand the experimental results in Fig. 2c, we must also take into account the cooperative effect of dense NP coverings (see Fig. 1d). Also, note that the experimental SERS intensity results from the combined scattering from target molecules on all NPs. Accounting for this in the theoretical modeling, we assume a NP is coupled to each NT unit cell and calculate the total conductance from the NT to the molecules (Methods). The result is shown in Fig. 5f, and demonstrates the same features as the experimental data (Fig. 2b, c)—a sharp electric-field threshold to catalytic activation, strongly enhanced intensity over a window of applied voltage, and then subsequent intensity decay on further increasing the voltage. To confirm this, we also implemented a more realistic model of the FF-PNT, taking into account the stacked layers of six-fold macrocycles (Fig. 4c and Methods), and very similar results are obtained—see inset to Fig. 5f. We again emphasize that the reduced quantum model is designed to capture the essential physics of the problem, not to provide quantitative agreement with experiment. Our theoretical results do demonstrate that spectral weight redistribution in the FF-PNT induced by an applied electric field does activate a facile charge-transport channel across the template, which in turn we argue is responsible for enhanced catalytic function for oxidation reactions of target molecules.

**Heating effects and temperature-gradient sensing**. The local heat generated from metal NPs due to LSPR excitation during laser illumination can enhance photocatalytic activity[11]. Heat generated from the FF-PNT/Ag NPs template with PATP was measured during the use of an electric field (Supplementary Fig. 17); we find that the temperature increases linearly with field strength. FF-PNTs are known to have piezo and pyroelectric characteristics that can be affected by temperature[10,11,15–17,29]. The increased temperature arising from an applied electric field may therefore lead to an enhanced pyroelectric effect for the FF-PNTs, where charge excited during heating can be transferred to Ag NPs[10,11,15–17,29,39,40]. It should be noted that the pyroelectric effect of FF-PNT decreases as the temperature increases above ~60 °C[15]. This is keeping with our measurements which show that SERS intensity decreases for voltages >~25 V (Fig. 2c), as the temperature reached ~70 °C.

Thermally-induced water loss and corresponding changes in the chemical and hydrogen bonds that hold the FF-PNT structure together may also play a role. The applied electric field introduces local heating that leads the water molecules present in the PNTs to evaporate[15]. Studies have shown that the increase in temperature could possibly effect the organization of water molecules in the nanochannel of individual nanotubes[15–17]. Reduction in SERS intensity observed after heating above 50 °C (Supplementary Fig. 18) can be as a result of the hydrophilic temperature-dependent binding of water with $COO^-$ tails, and the disturbance of dipole ordering in the tubes interior. The dipolar organization of water molecules positioned near carboxyl groups in the nanocavities cannot convalesce at these temperatures due to their high mobillity, resulting in a measured decrease of pyroelectric current and hence reduction in SERS signal[15–17]. Such temperature-dependence can be applied as a nanoscale thermometer amenable with a biological environment. An advantage of such pyroelectric thermometers based on FF-PNTs ($2 \mu C/(m^2 K)$ pyroelectric coefficient) is their sensitivity to the temperature gradient. This would be useful, for example, in monitoring fast thermal properties in cells. A second proposed use is in thermal energy harvesting, which may participate with piezoelectric harvesting when abrupt differences of temperature occur, such as in living human systems[15–17].

Our theoretical studies show that a necessary condition for photocatalysis is facile charge transfer across the template, activated at a critical threshold electric field. Hot electrons produced by photoexcitation of NPs and lost in the oxidation reaction of the molecule can thus be replaced, permitting a sustainable catalytic cycle. The plasmonic particles can generate new and alternate reaction conduits by the creation and injection of hot electrons and via charge transfer based processes between FF-PNTs and Ag NPs. However, we have also shown experimentally that heating effects are important, due to the known piezoelectric and pyroelectric properties of the FF-PNTs[15–17]. These likely contribute to the dramatic enhancement of catalytic activity in this FF-PNT/Ag NP template system.

**Oxidation of 2-AMP**. To demonstrate the versatility and robust functionality of the FF-PNT/Ag NP template, we now turn to the oxidation reaction, 2-AMP → 2-NIP. The Raman and SERS spectra for 2-AMP are well-characterized and are known to undergo photochemical change on oxidation to 2-NIP in the presence of plasmonic metals[25,27,41–44]. SERS measurements of 2-AMP on the FF-PNT/Ag NP template were recorded under the same conditions as for PATP. The applied electric field is shown to influence the spectral behavior of the analyte molecule—as well as SERS intensity (Fig. 6a–d and Supplementary Fig. 19), similar to the PATP case discussed above (Fig. 2).

Applying a voltage to produce an electric field, but keeping the offset bias at ≤100 mV, resulted in a (six-fold) increase in SERS signal intensity with no change in the SERS spectral relative band intensities (as for PATP). However, when the offset bias was raised above 100 mV the (eight-fold) SERS intensity increase was accompanied by significant changes in spectral relative band intensity. The strong appearance of bands at 1285 (C–N stretching mode) and 1334 (amine scissoring mode) $cm^{-1}$ signifies the transformation from 2-AMP to 2-NIP[41–44]. A lowering of the SERS Raman band frequency to $1334 \, cm^{-1}$ from $1342 \, cm^{-1}$, as well as intensity increases for bands at 1285 and $1334 \, cm^{-1}$ implies transformation from 2-AMP to 2-NIP through a strong chemical interaction of 2-AMP with the substrate through the nitro group[41–44]. In the absence of FF-PNTs, Ag NPs alone do not result in changes to the 2-AMP spectral features with an applied electric field (Supplementary Fig. 20). A plot of Raman band intensities (Fig. 6b) associated with 2-NIP (at 1258 and $1334 \, cm^{-1}$) with electric field shows that a threshold field strength is required for the formation of these two bands to in the SERS spectra. Note, however that further increasing the field causes the SERS signal associated with 2-NIP to decrease (Fig. 6c). Similar behavior was observed for PATP in Fig. 2. Removal of the electric field causes the relaxation of the SERS intensity and results in the back reaction 2-NIP → 2-AMP (Supplementary Fig. 21).

Combining both electric field with super band irradiation of the FF-PNT at $\lambda = 254$ nm (Supplementary Fig. 22) resulted in the rapid transformation of 2-AMP to 2-NIP with a nine-fold increase in SERS intensity. This result provides further evidence that combining both electric field and UV irradiation significantly enhances such photocatalytic processes.

**Enhancing SERS sensing**. Finally, we investigate the alternative application of this FF-PNT/Ag NP template for enhanced SERS sensing of biomolecules such as glucose and other nucleobases (Fig. 7). Such biomolecules have a low Raman cross-section and are therefore challenging to observe directly. Below, we demonstrate that the template dramatically enhances sensing when an electric field is applied. SERS measurements of glucose (Fig. 7a, and Supplementary Fig. 23) on the FF-PNT/Ag NP template

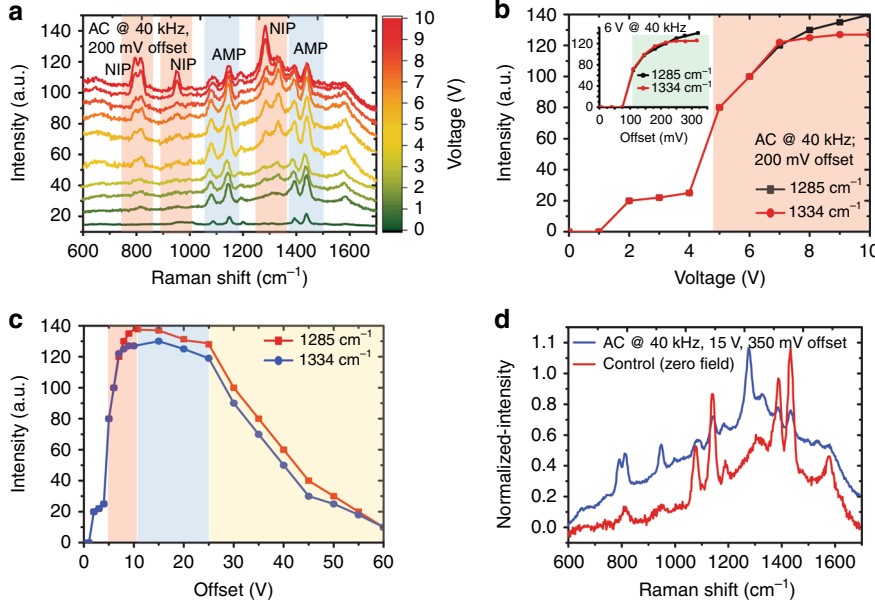

**Fig. 6** SERS study of 2-AMP on the template in an applied electric field. **a** SERS measurements of 2-AMP on the FF-PNT/Ag NP template at different electric fields strengths, applying voltages from 0–10 V at 40 kHz AC with offset bias of 200 mV. **b**, **c** Corresponding Raman band intensity of bands associated with 2-NIP as a function of voltage. **b** Critical threshold voltage ~4 V required for the formation of 2-NIP. Inset shows the SERS signal strength as a function of offset bias. **c** The window of 10–20 V is optimal for 2-NIP product yield via photocatalysis, with SERS intensity reducing above ~ 25 V. **d** Normalized SERS spectra, comparing a fully relaxed control sample at zero field (red) to the spectrum recorded with applied field (blue). The similarity with Fig. 2 demonstrates robust and versatile template functionality

showed Raman bands in good agreement with other reports[45,46]. With increasing electric field, strong and clear appearance of all glucose bands combined with an eight-fold increase in SERS intensity was seen. Relaxation of the Raman signal (Supplementary Fig. 23) occurred following removal of the electric field, as with previous probe molecules (PATP and 2-AMP). Uniformity and reproducibility tests were performed on the sample at different spots or different samples, with only a 5% change observed (Supplementary Fig. 23)

To further future demonstrate the applicability of FF-PNT/Ag NP templates with applied electric field, the nucleobases uracil and thymine were investigated, as well as the proteins bovine serum albumin (BSA) and albumin (Lys-egg and Lys-human)—see Fig. 7 (b–f) and Supplementary Fig. 23. In optimized conditions (300 mV offset bias, 10 V AC at 40 kHz) a seven to eight-fold intensification in SERS signal was observed (compared with zero-field measurements). The Raman bands recoded in our study are in line with previous reports[47–50] (Supplementary Tables 1 and 2). For all molecules, a return to the original Raman signal level occurred when turning off the electric field, as with previous probe molecules (PATP, 2-AMP, and glucose). Supplementary Fig. 24 further establishes the high degree of reproducibility of our results, taking BSA and Lys target molecules as examples. We tested with and without field, over different samples at different spot positions, finding only a 5–6% variability in signal intensity.

Since DNA-based molecules are notoriously difficult to detect due to their small Raman cross-sections[47–50], our results demonstrate the power and utility of employing the FF-PNT/Ag NP template with an applied field to enhance sensing. We note that SERS sensing of guanine, adenine, and cytosine target molecules was similarly enhanced. However, these molecules were also oxidized by the template in applied field (Supplementary Fig. 25).

## Discussion
Our results show that the FF-PNT/Ag NP template controls and greatly enhances product formation and selectivity in plasmon-activated catalytic reactions. Applying an external electric field of sufficient strength to the FF-PNT/Ag NP template, together with LSPR excitation, resulted in the catalyzed oxidation of PATP to PNTP. Interestingly, the reaction is fully reversible, with PNTP molecules reduced back to PATP after switching off the field. Similar results were obtained in a second probe molecule, 2-AMP. The FF-PNT/Ag NP template affects the plasmonic photocatalyst reaction by creating different reaction pathways through the formation and injection of LSPR-excited hot electrons and through charge transfer processes involving FF-PNTs and Ag NPs. Our detailed theoretical analysis, based on full quantum transport calculations, demonstrates that the key ingredient necessary for template functionality is a channel for facile charge redistribution across the template, here activated by electric field application. Both the pyroelectric and piezoelectric properties of FF-PNTs with an applied electric field may play a part in the improvement of the catalyst activity, by facilitating charge transfer processes across the template. In addition, heat generated from both Ag NPs and the FF-PNTs because of the absorbed Raman excitation photons or applied voltage may also change the reaction conditions. Furthermore, we have successfully detected a number of biomolecules such as glucose, thymine, and uracil, without any further chemical treatment or binder molecule, using the highly stable FF-PNT/Ag NP template with electric field, resulting in eight-fold increase in SERS intensity. Our results may enable a new platform technology for monitoring catalytic activity and surface-enhanced Raman scattering of biomolecules with low Raman cross-sections, utilizing the strength and frequency of the applied field. In addition, the capacity to dynamically control SERS spectra of specific bond types in situ adds an extra dimension to the collection of SERS measurements in nanocomposite structures based on biological materials such as FF-PNTs. We finally note that our approach is versatile. By matching the electronic bandwidth of plasmonic metal nanoparticles to the bandgap of 1D semiconductors, other material composites can be formed which are similarly expected to

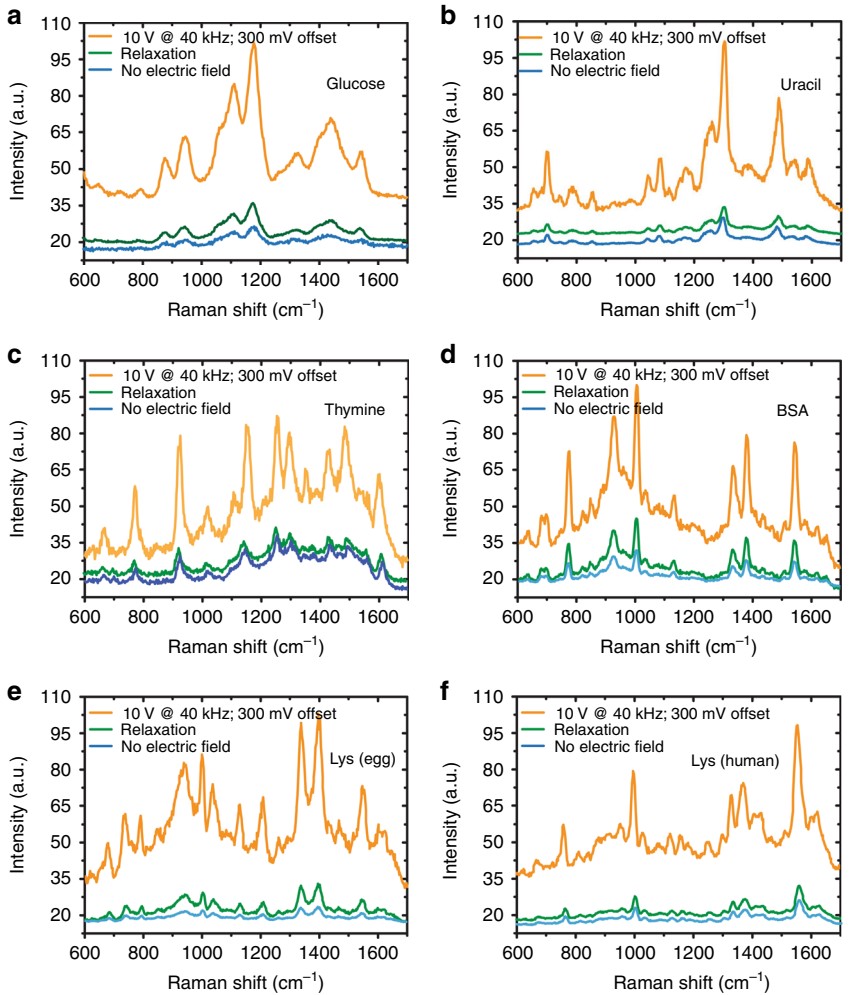

**Fig. 7** SERS sensing of biomolecules on the template using an externally applied electric field. SERS measurements of (**a**) glucose, (**b**, **c**) DNA based molecules uracil and thymine, (**d**) BSA, (**e**) Lys-egg and (**f**) Lys-human, demonstrating field-enhanced sensing using the FF-PNT/Ag NP template

enhance plasmonic catalysis and SERS sensing on application of an external electric field.

## Methods

**Preparation of the Si substrate with gold electrode pair**. Si wafers (Si Mat) were cut as $2 \times 1$ cm samples. The Si wafer surfaces where first cleaned by immersing in acetone for 2 min, followed by washing with ethanol and isopropanol. De-ionized water was then used to rinse the substrates, followed by the application of nitrogen gas to blow dry the silicon's surface of any residue solvent. To make the gold electrode pair on the Si substrate, a 3D printed mask with dimensions of $2 \times 1$ cm and opening 0.1 mm was used. Gold was sputtered through mask openings forming the electrode pair. The process is illustrated in Supplementary Fig. 1.

**Preparation of FF-PNT/Ag NP template**. The FF-PNTs were made by dissolving L-diphenylalanine peptide (Bachem) in the solvent 1,1,1,3,3,3-hexafluoro-2-pro-panol (Sigma-Aldrich). Using a starting concentration of 100 mg/ml, this been diluted in deionized water to a concentration of 2 mg/ml. The FF-PNT/Ag NP templates were prepared using 2 mg/ml of this solution heated at 100 °C for 2 min. Twenty microliters of Ag NP solution presenting a concentration of 0.02 mg/ml in water was added to 60 μl of the heated FF solution and stirred for 3 min. Following this 30 μl of the mixed solution was then placed on the gold electrode patterned Si substrate to create the aligned FF-PNT/Ag NP template. For the control sample; Ag NPs on a Si substrate. This sample was made using 20 μl of Ag NPs (0.02 mg/ml) which was diluted in 60 μl of water then 30 μl of the solution was deposited on the Si substrate. SEM images show the alignment of the FF-PNT/Ag NP template in the gap between the gold electrodes (Supplementary Fig. 13). The diameter of the FF-PNTs, determined from SEM images, was $3.2 \pm 1.3$ μm from $n = 30$ FF-PNTs.

**Probe molecule solutions**. 4-aminothiophenol (PATP) (New Star Chemical) and 2-aminothiophenol (2-AMP) (Sigma-Aldrich, Ireland) solutions were prepared in

methanol to a concentration of $10^{-4}$ M. These solutions where then diluted with deionized water to a final concentration of $10^{-5}$ or $10^{-6}$ M. Thymine cytosine, uracil, adenine, and guanine (all nucleotides sourced from Sigma-Aldrich) solutions were prepared in deionized water to a final concentration range of $10^{-5}$ M. Similarly, with glucose, TMPyP (5,10,15,20-Tetrakis(1-methyl-4-pyridinio)por-phyrin tetra(p-toluenesulfonate), BSA (Bovine Serum Albumin), Lys-egg and Lys-human (Albumin from chicken egg white and humans, respectively) all sourced from Sigma-Aldrich Ireland.

**Spectral characterization**. Optical absorbance measurements (V-650, JASCO, Inc.) where preformed using the following settings of a 1 nm step size, with a 1 nm bandwidth, and a 400 nm/min scan speed. Recording spectra over a 190–900 nm wavelength range. Optical absorbance measurements then were performed on the FF-PNT/Ag NP template that was aligned on cover slip with electrodes during electric field application with different AC voltages between 0 V and 10 V, frequency of 40 kHz, and offset bias of 5 to 60 V.

Fourier Transform Infrared Spectroscopy (FTIR) was undertaken using an Alpha Platinum Bruker system. To record FTIR spectra FF-PNT/Ag NP solution (10 μl) was deposited onto the ATR interface. Spectra were collected using transmission mode scanning from 1400–4000 nm.

A Scanning Electron Microscopy (SEM) (JSM-7600F) system was employed to characterize the samples. To obtain SEM images a thin (~8 nm) layer of gold was sputtered on the samples before SEM imaging (Hummer IV, Anatech USA).

To record SERS spectra a bespoke Raman system was used. This system comprised of an inverted optical microscope (IX71) with a SP-2300i spectrograph (Princeton Instruments), and a EMCCD camera (IXON). The 532 nm Raman excitation laser was fixed at a 5 mW incident power. SERS measurements were performed using an arbitrary waveform generator (Tektronix 3390) with different AC voltage (0 to 10 V) and frequency (from 0 to 100 kHz) and offsets of 0 to 500 mV. Offsets in the range of 5 to 60 V were applied using a DC power supply (TENMA, 72–2015). UV irradiation was applied using a UV pencil lamp with a

wavelength of 254 nm possessing an output power of 4.5 mW/cm² (Edmond Optics) when located at a distance of 2 cm from the sample.

**Application of applied electric field.** During SERS measurements, an electric field was applied. This was generated by applying a voltage across the electrode pair on either side of the FF-PNT/Ag NP template, at 0–10 V and 1–105 kHz, resulting in electric fields upto 10 V/mm. An Offset was also applied, ranging between 0 and 355 mV. AC frequency was fixed at 40 KHz which was found to produce the maximum enhancement on SERS enhancement and plasmonic catalysis (Supplementary Fig. 7). In contrast the applied voltage and also the applied offset bias both impacted the photochemical reaction yield and rate (as shown in Fig. 2, Supplementary Fig. 6).

**Theoretical calculations.** For the theoretical modeling described in this paper, we formulated and analyzed the following fully quantum mechanical Hamiltonian, $H = H_{NT} + H_{NP} + H_{mol} + H_{NT-NP} + H_{NP-mol} + H_{probe}$. Here, $H_{NT}$ and $H_{NP}$ describe the isolated NT (given by Eq. 1) and the isolated NP (Eq. 2), respectively. The isolated PATP molecule is likewise described by a minimal tight-binding model of the molecular backbone, with parameters chosen to reproduce the HOMO–LUMO gap extracted from the experiment. These three basic elements of the full system are coupled together by,

$$H_{NT-NP} = V_{NT-NP}\left(c_{0,a}^{\dagger} d_{\vec{r}_1} + \text{H.c.}\right),$$

$$H_{NP-mol} = V_{NP-mol}\left(d_{\vec{r}_2}^{\dagger} a_{FO} + \text{H.c.}\right), \tag{5}$$

which describe local electronic tunneling between the NT, NP and molecule. Here, we assume the NP is attached to the middle of the NT (at unit cell $n = 0$). For NPs with $N = 75$ atoms across the diameter, we use $\vec{r}_1 = (38,38,1)$ and $\vec{r}_2 = (38,38,75)$, corresponding to NT and molecule attachment at the center of opposite NP faces. $a_{FO}$ is an operator for the molecule frontier orbital coupling to the NP. We set $V_{NT-NP} = V_{NP-mol} = 1$ for simplicity. To quantify the charge flow across the template into the molecule, we also introduce into the model a term $H_{probe}$, describing a metallic probe electrode contacting the molecule, which acts as a reservoir into which charge is absorbed (physically equivalent to an STM tip). The current $I$ into this probe electrode, due to an infinitesimal test bias, defines the calculated conductance $G = dI/dV\,|_{V \to 0}$ in linear response. The probe electrode and its coupling to the molecule are characterized by the hybridization $\Gamma$ featuring in Eq. 4. The coupling of the probe electrode to the molecule is assumed to be very weak so that properties of the system itself are not perturbed by it, $\Gamma \ll 1$ (the STM limit). Results for the conductance are presented in units of $\Gamma$.

The desired conductance is calculated exactly[34–36] via Eqs. 3 and 4, and requires a knowledge of the free NT Green's function $\left[\mathbf{G}_{NT}^0\right]_{ij} \equiv \left\langle\left\langle c_{0,i}; c_{0,j}^{\dagger}\right\rangle\right\rangle^0$ computed from $H_{NT}$, and the trans-template propagator $\mathcal{G}_{NT-mol} = \left\langle\left\langle d_{\vec{r}_1}; a_{FO}^{\dagger}\right\rangle\right\rangle$ of the full system, $H$. We first obtain $\mathbf{G}_{NT}^0$ using (recursive) equations of motion[37], whose exact solution is a generalized matrix continued fraction,

$$\left[\mathbf{G}_{NT}^0\right]^{-1} = \mathbf{E}_0 - \mathbf{M}\left[\mathbf{E}_1 - \mathbf{M}[\mathbf{E}_2 - \mathbf{M}[\dots]^{-1}\tilde{\mathbf{M}}]^{-1}\tilde{\mathbf{M}}\right]^{-1}$$
$$\tilde{\mathbf{M}} - \tilde{\mathbf{M}}\left[\mathbf{E}_{-1} - \tilde{\mathbf{M}}[\mathbf{E}_{-2} - \tilde{\mathbf{M}}[\dots]^{-1}\mathbf{M}]^{-1}\mathbf{M}\right]^{-1}\mathbf{M} \tag{6}$$

where in this case $[\mathbf{M}]_{ij} = \delta_{ia}\delta_{jb}(t - \Delta)$, and $[\mathbf{E}_n]_{aa} = [\mathbf{E}_n]_{bb} = \omega + i0^+ - n\delta V$, and $[\mathbf{E}_n]_{ab} = [\mathbf{E}_n]_{ba} = t + \Delta$. All matrices are $2 \times 2$ since there are two orbitals in the NT model unit cell. To further substantiate our results, we also implemented a more sophisticated model of the real FF-PNT, comprising stacked FF macrocyle rings (Fig. 4). The FF unit consists of 6 diphenylalanine units coupled in a ring. We took into account the 48 backbone C, N, O atoms in the FF macrocycle (at the level of one active orbital involved in transport per atom), with the structural information and coupling strengths encoded in $\mathbf{E}_n$. Stacked FF rings forming a quasi-1d tube are coupled together by C=O…H–N hydrogen bonds, as encoded in $\mathbf{M}$. FF-PNT Green's functions for this model are obtained using the above equation, Eq. 6, apart from using these new $48 \times 48$ matrices $\mathbf{E}_n$ and $\mathbf{M}$. In this model, to reproduce the experimental semiconductor bandgap of 4.6 eV, we took all C–C and C–N single bonds to have equal coupling strengths $t_{sb} = 4$, while the C=O double bond was taken to be $t_{db} = 8$, and the hydrogen bonds $t_{Hb} = 0.6$. The final results using the two NT models are compared in Fig. 2f and are qualitatively equivalent.

The full $\mathcal{G}_{NT-mol}$ is obtained by coupling together the isolated NT, NP, and molecule. This can be done efficiently using the T-matrix formalism in real space[33],

$$\mathcal{G}_{xy} = \mathcal{G}_{xy}^0 + \sum_{p,q} \mathcal{G}_{xp}^0 T_{pq} \mathcal{G}_{qy}^0,$$

$$\mathbf{T} = \left[\mathbf{I} - \mathbf{H}_1 \mathbf{G}^0\right]^{-1}\mathbf{H}_1, \tag{7}$$

where the 0 superscript indicates Green's functions of the uncoupled system as before, $[\mathbf{T}]_{pq} \equiv T_{pq}$ is the T-matrix describing electronic scattering between sites $p$ and $q$, which are the specific sites coupled by $\mathbf{H}_1$. Here, $\mathbf{H}_1$ is the matrix representation of $H_{NT-NP}$ or $H_{NP-mol}$ in the full Hilbert space of $H$. Eq. 7 is an exact

non-perturbative expression, and can be used successively to couple the molecule to the NP, and then to couple this system to the NP. To do this, we require electronic propagators across the NP, $\left\langle\left\langle d_{\vec{r}_1}; d_{\vec{r}_2}^{\dagger}\right\rangle\right\rangle^0$. In practice, this can be done accurately and efficiently using the convolution method introduced in ref. [38].

Finally, we discuss a straightforward modification to the above protocol to make better connection with the experiment. Instead of considering a single NP located at the middle of the NT, we now generalize to a dense covering of NPs. Specifically, we take one NP per unit cell of the NT (clearly an approximation, but one which recovers the essential physics of the problem). The effect of the hybridization between NT and NP at each unit cell can be simply incorporated on the level of the NT Green's functions by modifying $\mathbf{E}_n$. The dense NP coverage has a cooperative effect on the NT density of states, and therefore the conductance through each NP. Denoting the conductance between the NT and a target molecule attached to a NP at site $n$ as $G_n$, we now compute the total conductance as $G_{total} = \sum_n G_n$. This is plotted in Fig. 5f, and captures the basic features of the experiment, including the field-activated enhancement and threshold behavior.

## Data availability

All the data are available from the authors.

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

## Acknowledgements

This research was funded by the Ministry of Higher Education of Saudi Arabia under the King Abdullah Scholarship Program (ref. no. IR10161), the European Union's Horizon 2020 research and innovation program under Marie Skłodowska-Curie grant agreement number 644175, and Science Foundation Ireland (12/IP/1556 and 07/IN1/B931). A.K.M. acknowledges funding through the Irish Research Council Laureate Award Programme 2017/2018 (IRCLA/2017/169). The authors acknowledge Ian Reid for assistance with SEM and XPS, the UCD Centre for BioNano Interactions for access to the fluorescence spectrophotometer, Gareth Redmond for access to UV-vis, and Aaron Martin for access to FTIR. The authors acknowledge Michael McDermott and Jim McDaid in the School of Physics electronics workshop for use of a voltmeter with thermocouple sensor and a DC power supply. We also thank to Jose Lopez for 3d printing the mask used to deposit interdigitated electrodes.

## Author contributions

S.A. carried out the experiments and processed data. S.A., B.R., and J.R. designed the experiments and developed the experimental setup. S.A. carried out sample preparation, Raman measurements, FTIR, UV-vis measurements, SEM, and optical imaging. S.T.B. performed IV measurements. A.K.M. carried out all the theoretical work and calculations. S.A., B.R., J.R., and A.K.M. wrote the manuscript. All authors discussed results and reviewed the manuscript.

## Additional information

**Competing interests:** The authors declare that they have no competing interests.

**Journal Peer Review Information:** *Nature Communications* thanks Pedro Camargo and the other, anonymous, reviewer(s) for their contribution to the peer review of this work. Peer reviewer reports are available.

