## [Peer Review File · Nature Communications]

Reviewers' comments:

Reviewer #1 (Remarks to the Author):

This paper represents an important contribution in the field of SERS detection and plasmonic enabled transformations. The authors employ FF-PNT nanotubes decorated with Ag NPs as substrates and demonstrate that the selectivity and efficiency in plasmonic mediated transformations can be tuned by the application of an external electric field. Moreover, this approach can also be put to work toward the SERS detection of challenging Raman probe molecules. The authors also developed a model to explain their results and the effect of the external electric fields. This paper is suitable for publication after addressing the following issues:

- please revise the text one more time. There are a few typos along the ms.
- references should be included in the first paragraph of the introduction.
- Why the use of FF-PNTs relative to other supports or semiconducting materials? Is this effect also expected? How versatile would be the approach?
- Please discuss the size of the Ag NPs, include size histograms, and comment on the importance of size vs. plasmonic effects and the generation of charge carriers vs scattering under LSPR excitation.
- include in the main text the wavelength employed in the Raman measurements
- Does the substrate need to be aligned?
- under 532 nm, can resonance Raman effects be observed in addition to SERS? Could this be contributing to the detected signals? Did the authors try to perform measurements under excitation at longer wavelengths?

Reviewer #2 (Remarks to the Author):

The author reports a template comprised of silver nanoparticles on self-assembling diphenylalanine peptide nanotubes that could be employed for field-activated enhanced photocatalysis and biomolecular sensing. The following comments need to be addressed before it can be considered for publication.

1. The author should highlight and focus on the novelty of this paper. If the author claims that the novelty of this paper is in the field-controlled plasmonic photocatalysis and the first group to observe this process, please explain how findings in this work are different from that reported by other groups [1-3] in the literature. Otherwise, if the novelty is that field-controlled plasmonic photocatalysis has not been demonstrated in FF-PNT/Ag NP, publication in nature communication would be feasible only if the enhanced Raman properties and plasmonic photocatalysis properties of FF-PNT/Ag NP have not been reported in the past [4, 5] by your group.

2. The author suggests that the LSPR-excited hot electrons and electrons transferred from FF-PNTs to Ag NPs contribute to the O₂ activation step, leading to the formation of PNTP. This formation leads to the change in the SERS spectra. Please explain if this hypothesis also applies to the variation in Raman modes noted in Fig. 7. For example, some of these Raman peaks seem to be enhanced, or suppressed in the presence of electric field (circled in red). Explanations should be given to the readers if these different modes were also due to the some catalytic processes, or some other types of chemical transitions. Clarifications should be given to distinguish and justify the origin(s) of the changes in the Raman modes. Otherwise, this hypothesis for the formation PNTP may be difficult to convince the reader.

3. In the manuscript, the author stated that the hot electrons are transferred to adsorbed O₂ molecules, generating triplet diradical ³O₂ formed in degenerate antibonding π_{2px} and π_{2py} orbitals, that subsequently is involved in the oxidation of PATP to DMAB. Please comment on the field-assisted splitting of water molecules possibility due to the small gaps [6] formed from these FF-PNT/Ag NP, which could also assist the oxidation process. These small gaps in the FF-PNT/Ag NP could lead to intensively high electric field, even though the applied voltage could be low. In addition to the variation in O₂ concentration, since the system could be viewed as an integration of numerous nano/micro- scaled electrolysis cells, please also discuss on the variations of concentration in different components (that is PATP, PNTP, DMAD, 2-AMP, and 2-NIP) in terms of electroreduction as well.

4. Eight-fold increase in SERS intensity is observed with the FF-PNT/Ag NP template with electric field in comparison with that in the absence of an electric field. The author attributed this increase to the better charge transfer of the LSPR-excited hot electrons. Discussion should also be included about the aggregation and variation in the geometrical topologies and morphologies of the Ag nanoparticles and FF-PNT/Ag NP template under the influence of electric field, as the SERS measurements are performed on analyte solutions. These changes will affect the extinction coefficient (not just based on absorbance alone) which could lead to the difference/augmentation in the SERS enhancements.

5. Figure S1 seems to be misleading to the reader. In Fig. S1c, the area of Si substrate deposited with FF-PNT/Ag NP shows aligned lines for locations without UV/ozone exposure. This gives the impression to the reader that FF-PNTs could be aligned on Si substrate even without exposure to UV/ozone. Maybe the author could consider demarcating the areas where the FF-PNTs are aligned/randomly orientated.

References

- [1] D. A. Nelson, and Z. D. Schultz, "Influence of optically rectified electric fields on the plasmonic photocatalysis of 4 - Nitrothiophenol and 4 - Aminothiophenol to 4,4-Dimercaptoazobenzene," *The journal of physical chemistry C*, 122, 8581 (2018).
- [2] M. Sun, Z. Zhang, H. Zheng *et al.*, "In-situ plasmon-driven chemical reactions revealed by high vacuum tip-enhanced Raman spectroscopy," *Scientific Report*, 2, 647 (2012).
- [3] L. B. Zhao, J. L. Chen, M. Zhang *et al.*, "Theoretical study on electroreduction of p-nitrothiophenol on silver and gold electrode surfaces," *J. Phys. Chem. C*, 119, 4949 (2015).
- [4] S. Almohammed, S. Fedele¹, B. J. Rodriguez *et al.*, "Aligned diphenylalanine nanotube - silver nanoparticle templates for high - sensitivity surface - enhanced Raman scattering," *Journal of Raman Spectroscopy*, 48, 1799 (2017).
- [5] S. Almohammed, F. Zhang, B. J. Rodriguez *et al.*, "Photo-induced surface-enhanced Raman spectroscopy from a diphenylalanine peptide nanotubemetal nanoparticle template," *Scientific Report*, 2018(8), 3880 (2018).
- [6] Y. Wang, S. R. Narayanan, and W. Wu, "Field-assisted splitting of pure water based on deep-sub-debye-length nanogap electrochemical cells," *ACS Nano*, 11, 8421 (2017).

Reviewer #3 (Remarks to the Author):

The authors propose a new platform for photocatalysis and SERS-based sensing through a controllable charge transfer process following a semiconductor (FF-PNT)-metal-molecule routine. The electrons from the FF-PNT semiconductor substrate can be transferred into the plasmonic Ag nanoparticles attached on the PNT surface, activating the photocatalysis (oxidation) for the molecules adsorbed on the metal surface. Meanwhile, the same charge transfer can enhance the amplitude of SERS signals by a well-known "chemical enhancement mechanism" that have been discussed for decades. The authors have extensive experiments on the impacts of the AC frequency and amplitude, DC amplitude, different combination of template/device substrates, and atmospheric changes (e.g. UV excitation and heating). In addition, there is an initial physical modelling work to explain the trend of the interaction inside the described platform.

1. Overall, it's an interesting idea to select a bandgap matching semiconductor material and integrate it with the plasmonic reactor/sensor platform. Not like the conventional metal/TiO₂ or similar hybrid substrate, the described FF-PNT-AgNP substrate provides a new way of electrical control on the reaction or sensing. Although I think this work should be interesting to Nature Communications readership, I still have a bunch of questions and comments, which should be better addressed by the authors before I would further recommend publication.

2. Detailed questions and comments:

3. Electrochemical SERS (EC-SERS) and catalysis are very common technologies in these days, why should we use FF-PNT or semiconductor-plasmonic hybrid platform? We would directly combine TiO₂ like photon-active semiconductor with plasmonic materials to gain a simple use case with only light illumination to active catalysis and strong sensing, etc. The describe FF-PNT-AgNP platform actually needs both light illumination, a high AC voltage (10V) and a DC voltage. Perhaps it may have a better control, but its use is very complex. We usually develop a tailored technology to facilitate the use, but not complicate it. In an EC-SERS, a simple and small bias works for both sensing and catalysis. I don't

really get the point of authors' claim on "our results point in the direction of future rational design of templates tailored for specific use".

4. What's EL-SERS? What's the abbreviation of EL? Is it different to the electrochemical SERS?

5. I find it's very strange to use the Si substrate, especially without a thick electrical passivation layer, e.g. SiO₂. Although I'm not very sure, it seems the two Au electrodes are shorted by the Si substrate. The resistance of the FF-PNT layer is similar to that of the Si substrate from the resistances shown by the authors. The SiO₂ passivation layer only appeared at the gap of electrodes and was formed by UV/Ozone. What's the thickness of the SiO₂? Does it thick enough to avoid the electric leakage?

6. How AgNPs disperse on the surface with or without FF-PNT is not that clear? From the SEM pictures in the manuscript, they were very isolated from each other, meaning no current at biased AC/DC voltage. However, in Fig. S9b, the DC offset did increase the intensity of the PATP Raman. Why this can happen? Was there some extensive aggregations of AgNPs, which may form a kind of electrically conductive network?

7. This comes to my biggest question on this manuscript: were the measurements done in air after drying of the added sample droplet or directly in a sample solution? This important detail was not mentioned at all.

8. Another question related to the use of Si substrate. Si can absorb the excitation photons and sequentially forms carriers, which may have an impact on the electrochemistry at the Ag NPs. Have the authors been aware of this issue?

9. Why use an intense AC voltage together with a weak DC offset? From the result, the DC voltage is used for the oxidation. In the method section, the authors said the frequency has a small impact on the photochemical reaction. What's the role for the AC voltage? Will it still work by adding a high DC voltage like 4V? It should be explained somewhere in the content.

10. In the frequency testing of AC voltage, such as Fig. S6a and Fig. S14a, the DC offset was 1 mV and the AC was 1V. I can understand this is a control experiment. But why not do the frequency testing at the typical working parameter, such as AC 6 V, DC 300mV? How does the frequency of AC voltage influence the reaction/biosensing with this setting? I think this is a more important control experiment, which should be added in the manuscript.

11. The authors have ascendingly scanned AC voltages from 0 to 60 V. How about descending scan from 60 V back to 0 V? And how about the DC scan? It may indicate some charge trapping in the FF-PNT-AgNP-molecule system.

12. How many devices have been tested? I suppose every measurement of a new sample needs a new device/template to avoid any possible cross-talk contamination. Can the self-alignment technology of FF-PNT and the Ag NPs coating always give the same or similar resistance? What's the reproducibility and the uniformity of such processing? How about the device/template variation in the entire work? Is the minimum required 100mV DC offset always the same for every device/template in use?

13. I'm not well convinced by the explanation of the electric field induced charge transfer between NT-NP-molecule. Electric field is always there, no matter the presence of FF-PNP. But the experimental observations were highly related to the use of FF-PNT. Should it be related to the local electrostatic potential? The simulation was a 1D configuration, and the potential on the PNT-AgNP was fixed. In practice, the template/device was a 2D or even 3D FF-PNT-AgNP-molecule. The electrostatic potential

should change from one electrode to the other, even in the presence of both AC and DC voltages. I wonder, have the authors studied different locations? Were the threshold AC (~4V) or DC voltages (~100mV) dependent on various locations within the electrode gap? Were the SERS signals the same from various locations? Indeed, I have been aware the authors mentioned: "Uniformity and reproducibility tests were performed on the sample at different spots, with only a 5% change observed." In the section of Enhancing SERS sensing. But since this work not only discusses SERS enhancement, but also photocatalysis, AC/DC threshold, etc. Perhaps, the authors should give more details on this part.

14. In the theory described in Fig. 3, electrons can come from Ag NP and FF-PNT. To match the HOMO-LUMO gap and the Fermi level of Ag NPs, it requires an excitation at 532 nm. The same laser was used for Raman, and it became difficult to define the contributions from plasmonic hot electrons or electrons propagating inside the FF-PNT at the bias. Will it be possible to use a 785 nm excitation to mismatch the energy levels, avoid the plasmonic related charge transfer and only leave the contribution of the PNT. And 785nm can still support SERS for monitoring the reaction/sensing?

15. Clearly, the sample molecule PATP or PAMP can be oxidized in the described device/template. How about the FF-PNT? It's made by peptides. Will it be irreversibly damaged by triplet oxygen? The same question for the AgNPs in use. Is there any alternative solid-state semiconductor can be used to replace the organic semiconductor in the describe concept?

16. Fig. S11. The UV-vis absorption spectra. Why cannot see the LSPR resonance (~400nm) of Ag NPs in FF-PNT/AgNP spectra in both (a) and (b). BTW, the different orders of the major and the insert are annoying for reading.

17. I'm not very family with the modelling part. The Conductance of electrons in the NT-NP-molecule route sounds like the minimum physics to describe the charge transfer. I'll need to rely on other reviewers to evaluate the simplification for supporting a valid physical model.

18. A small comment: vibrational Stark effect can shift the Raman peak position. However, it usually requires a very high electric field (> 1 MV/m), which is not the case in the described work. It may have another explanation.

19. Some small typo mistakes:

P19, "DC offsets in the range of 5 to 60 V..." should be "AC offsets..."

What's the distance between the two electrodes: 0.1 or 1 mm. The main content says 1 mm, but the SM says 0.1 mm.

“Enhanced photocatalysis and biomolecular sensing with field-activated nanotube-nanoparticle templates”

Below we give our detailed response to each of the reviewers (reviewer comments in blue):

Reply to Reviewer #1

We thank the reviewer for their careful reading of the manuscript and insightful suggestions for strengthening the work. We are grateful, in particular, for the positive assessment and recommendation for publication:

“This paper represents an important contribution in the field of SERS detection and plasmonic enabled transformations... This paper is suitable for publication after addressing the following issues”

We reply to each of the substantive issues raised by the reviewer below, and have revised the manuscript accordingly.

- 1. Please revise the text one more time. There are a few typos along the MS*

We have revised the manuscript and thoroughly edited for typos and readability.

- 2. References should be included in the first paragraph of the introduction.*

We have reformulated the introductory paragraphs somewhat, now adding in complete references.

- 3. Why the use of FF-PNTs relative to other supports or semiconducting materials? Is this effect also expected? How versatile would be the approach?*

This is an excellent question that goes to the heart of our novel approach. Peptide nanotubes (PNTs) have many attractive advantages in comparison to other semiconductor materials -- including piezoelectric and pyroelectric effects, a wide band gap, robust and rigid structure, while also being biocompatible. These materials are also very easy to make (which has practical implications for scalable use) and they can be incorporated with metal nanoparticles (NPs) by simple mixing processes.

Importantly, the electronic states of peptide nanotubes are complementary to those of the Ag NPs, which turns out to be critical for the functionality uncovered in our work. In particular, the band gap of the peptide nanotubes is closely matched to the bandwidth of the metallic nanoparticles. The plasmonic properties of the NPs are therefore preserved. However, the application of an electric field along the length of the PNTs allows for conductive behaviour. Indeed, we show that charge transfer across the template -- from electrodes to PNTs to metallic NPs to target molecules -- enables catalytic oxidation reactions and enhanced sensing. This is also controllable in-situ by simply manipulating voltages. A key component to the experimental design is the 1D nature of the peptide nanotubes which are formed aligned (on SiO₂), which maximises the above electrical field effect on the template. These novel features are probed both experimentally and modelled theoretically to understand their role.

This is discussed more clearly in the revised manuscript. Furthermore, the introduction text has been modified to make more clear the importance and utility of peptide nanotubes:

“Nanocomposites combining semiconductor materials with metal NPs bring additional benefits to plasmonic photocatalysis. The contact potential difference between metal NPs and a semiconductor can separate photogenerated electrons and holes, thereby reducing electron-hole pair diffusion lengths and leading to more efficient photogenerated charge separation and transfer, which in turn enhances photocatalytic activity. Here we increase the probability and efficiency of both chemical reactions and SERS detection through electro-optical synergy, using a microfabricated chip design (in air rather than electrochemical). This is achieved through the use of a plasmonic-semiconductor system based on aligned diphenylalanine peptide nanotube wide band gap semiconductors (FF-PNTs). Diphenylalanine (FF), a short peptide consisting of two naturally occurring amino acid phenylalanine, can self-assemble into micro and nanosized tubular structures. The resulting organic self-assembled nanotubes are the FF-PNTs, which are stable/robust, rigid, and also biocompatible. They can be used in applications requiring the use of a wide bandgap semiconductor. FF-PNTs have been reported to have high thermal and chemical stability in addition to piezoelectric and pyroelectric properties. We show experimentally and theoretically that applying a longitudinal electric field allows the FF-PNT density of states to be tuned from a semiconductor to a metal, enabling effective charge transfer from the nanotube to the metal nanoparticles. This results in an enhancement in the state density of hot electrons. The effect is optimized through the physical alignment of the FF-PNTs, since the inherent electric dipoles of the FF-PNTs are then aligned and maximally responsive to the applied longitudinal electric field (a cooperative effect). We demonstrate that this optoelectrical device enhances photocatalytic conversion for model oxidation reactions (exemplified here by p-aminothiophenol (PATP) oxidized to p-nitrothiophenol (PNTP), and 2-aminophenol (2-AMP) oxidized to 2-nitrophenol (2-NIP)), exploiting the facile field-activated trans-template charge transfer. We also demonstrate that this same approach can be used to enhance the strength of Raman scattering from low Raman cross-section molecules such as glucose and DNA-based molecules, establishing the potential of our template design for sensitive detection and analytics. This approach is versatile and can be applied to a range of plasmonic metal nanoparticle and semiconductor combinations.”

As highlighted above, our approach is versatile. Theoretical simulations show that the important characteristics for successful operation is the matching of the NP bandwidth to the bandgap of a 1D semiconductor. For example, Ag or Cu NPs could be combined with organic PVDF (Polyvinylidene fluoride) or MgS 1D nanostructures; or Au NPs combined with AlN or ZrO₂ 1D nanostructures. However, we have found that the FF-PNT/Ag NP template is particularly well suited and optimized to purpose. It is also stable and cost effective.

We have revised the manuscript to include information demonstrating clearly the versatility of our approach. This point is also emphasized in the conclusion:

“We finally note that our approach is versatile. By matching the electronic bandwidth of plasmonic metal nanoparticles to the bandgap of 1D semiconductors, other material composites can be formed which are similarly expected to enhance plasmonic catalysis and SERS sensing on application of an external electric field”.

4. *Please discuss the size of the Ag NPs, include size histograms, and comment on the importance of size vs. plasmonic effects and the generation of charge carriers vs scattering under LSPR excitation.*

The requested information on effect of NP size was collected and studied but previously not presented (we simply use a representative NP size for the main paper and accompanying discussion). Now we include the full information as an additional figure in the Supplementary Information (see Supplementary Figure S16). This new figure shows data in agreement with the literature that larger Ag NP size results in higher SERS intensity, due to the greater electromagnetic field generated. Additionally, larger Ag NPs tend to aggregate, and this leads to more hotspots in the substrate. Again, this supports our observation in Fig. S16 that larger NP size leads to higher SERS signal and hence improved sensitivity of detection. We note that this result is in line with studies such as J. Phys. Chem. C 2011, 115, 5, 1403-1409: as nanoparticle size increases, scattering efficiency (SERS) *increases*. However, studies on the efficiency of hot-electron production (such as ACS Energy Lett. 2017, 2, 7, 1641-1653) report that as nanoparticle size increases the efficiency *decreases*. This study indicates that smaller Ag nanoparticles will produce a greater plasmonic catalysis reaction efficiency. Therefore there is a balance to be struck: greater SERS sensitivity favours large nanoparticle size, while greater catalytic activity favours smaller nanoparticle size. However, we note that in practice the 60nm nanoparticle sizes used for this study do still produce significant hot electron yield and hence enhanced catalysis while also retaining good SERS sensitivity. We have included a comment on this in the revised paper.

5. *Does the substrate need to be aligned?*

The FF-PNTs are essentially 1D semiconductors, which have particular electronic properties when an electric field is applied longitudinally: the varying potential down the NT activates facile charge transport when the field strength exceeds a critical threshold value, as shown explicitly by our theoretical modelling and also from the onset behaviour observed in our experiments. This is due to the inherent electric dipoles comprising the FF-PNTs. The physical alignment of an array of FF-PNTs is therefore necessary so that application of a electric field can be longitudinally aligned to produce the desired effect. We now clarify this important point directly in the revised manuscript (see also reply to comment #3 above):

“The effect is optimized through the physical alignment of the FF-PNTs, since the inherent electric dipoles of the FF-PNTs are then aligned and maximally responsive to the applied longitudinal electric field (a cooperative effect).”

6. *Under 532 nm, can resonance Raman effects be observed in addition to SERS?*

Yes. We do observe resonance Raman effects for particular molecules, such as the porphyrin TMPyP and methylene blue, which happen to be in resonance with the laser excitation wavelength (532 nm). This is now included in the main paper in Fig. 7 (panels g & h).

7. *Could this be contributing to the detected signals? Did the authors try to perform measurements under excitation at longer wavelengths?*

While additional resonance effects are observed for molecules in resonance with the excitation laser wavelength, most of the results presented are for molecules which are not in resonance. In all cases – whether resonant or off-resonant – we observe enhancement of photocatalysis and enhanced Raman sensing, indicating that the resonance phenomenon is incidental. This is now shown explicitly in Supplementary Figure S27. Note that we kept the laser excitation wavelength fixed, varying instead the probe molecules to examine resonance effects.

We also note the comprehensive set of control experiments performed to establish the robust nature of our results and the strength of our conclusions – see Supplementary Figures S9-S18.

Reply to Reviewer #2

We thank the reviewer for their careful reading of the paper and helpful constructive suggestions for improvements of the manuscript, which we have followed in full. Our response to specific queries is given below.

- 1. The authors should highlight and focus on the novelty of this paper. If the author claims that the novelty of this paper is in the field-controlled plasmonic photocatalysis and the first group to observe this process, please explain how findings in this work are different from that reported by other groups [1-3] in the literature. Otherwise, if the novelty is that field-controlled plasmonic photocatalysis has not been demonstrated in FF-PNT/Ag NP, publication in nature communication would be feasible only if the enhanced Raman properties and plasmonic photocatalysis properties of FF-PNT/Ag NP have not been reported in the past [4, 5] by your group.*

We are happy to clarify the novelty and originality of our work, and its relation to previous studies. We agree this is important to spell out clearly, and we have endeavoured to do that in the revised manuscript.

In particular, the referee outlined a number of results from other groups:

- [1] D. A. Nelson, and Z. D. Schultz, "Influence of optically rectified electric fields on the plasmonic photocatalysis of 4-Nitrothiophenol and 4-Aminothiophenol to 4,4-Dimercaptoazobenzene," *The journal of physical chemistry C*, 122, 8581 (2018).
- [2] M. Sun, Z. Zhang, H. Zheng *et al.*, "In-situ plasmon-driven chemical reactions revealed by high vacuum tip-enhanced Raman spectroscopy," *Scientific Reports*, 2, 647 (2012).
- [3] L. B. Zhao, J. L. Chen, M. Zhang *et al.*, "Theoretical study on electroreduction of p-nitrothiophenol on silver and gold electrode surfaces," *J. Phys. Chem. C*, 119, 4949 (2015).

This body of work shows the detailed understanding of plasmonic catalysis reactions that **electrochemical** SERS (with related theoretical studies supporting and deepening this understanding) and tip-enhanced Raman have provided.

Our work is distinct from previous works and is novel. First of all, we wish to emphasize right away that our work is not based on electrochemical SERS studies (as e.g. [1,3] above); all experiments were performed in a dry environment. We are also not in high vacuum or using a scanning probe, as per [2] above.

We specifically focus on the use of the complementary properties of a carefully chosen 1D semiconductor (peptide nanotubes) and plasmonic metal (Ag NPs) combination. Theoretical simulations and calculations (Fig 4 and 5 in paper) are the first of their kind for these kinds of systems and give a new and deeper insight. They demonstrate that the semiconductor band gap and NP density of states bandwidth should be approximately matched to achieve the desired field-controlled functionality. The semiconductor should be 1D – this is important because an applied electric field aligned with the tube activates metallicity above a threshold voltage, resulting in facile trans-template charge redistribution (the key ingredient for the photocatalysis and enhanced detection reported in this work). Our

peptide nanotube / Ag NP template achieves these specific conditions experimentally and is shown to operate efficiently. This is a new design paradigm.

Our field-activated approach to enhancing SERS and plasmonic catalysis reaction efficiency has not been previously discussed. As opposed to more standard UV activation techniques, our field-activated template is more efficient than Ag NPs or TiO₂ NPs without the PNT. Electric field-activation is also more responsive and allows more control over product formation (the reaction being fully reversible). Our 1d template design is also optimized by alignment of the PNTs in the field direction.

Furthermore, this new approach can be applied to alternative material composites as yet unstudied. We highlight this in the revised conclusion: "Through a rational design approach based on new theoretical insights, we match the density of states characteristics of the plasmonic metal nanoparticles with the band gap energy of a 1D semiconductor, demonstrating that the resulting template enhances SERS sensing and plasmonic catalysis reactions upon electric field activation. We note that other material composites satisfying the above criteria should provide similarly efficient functionality."

As an example of the latter, Ag or Cu NPs could be combined with organic PVDF (Polyvinylidene fluoride) or MgS 1D nanostructures; or Au NPs combined with AlN or ZrO₂ 1D nanostructures. However, we have found that the FF-PNT/Ag NP template is particularly well suited and optimized to purpose. It is also stable and cost effective.

Previous theoretical studies (e.g. [3] mentioned by the Reviewer) have not tackled the issue of trans-template charge transfer in full quantum mechanical simulations of this type. Here we use rigorous Green's function and T-matrix methods, combined with effective tight-binding quantum mechanical models of each template component, and the coupling between them (nanotube, nanoparticle, and molecule). This new approach yielded new insights, as described above.

In the main test of the paper itself, we also clarify the novelty of our approach:

"Nanocomposites combining semiconductor materials with metal NPs bring additional benefits to plasmonic photocatalysis. The contact potential difference between metal NPs and a semiconductor can separate photogenerated electrons and holes, thereby reducing electron-hole pair diffusion lengths and leading to more efficient photogenerated charge separation and transfer, which in turn enhances photocatalytic activity. Here we increase the probability and efficiency of both chemical reactions and SERS detection through electro-optical synergy, using a microfabricated chip design (in air rather than electrochemical). This is achieved through the use of a plasmonic-semiconductor system based on aligned diphenylalanine peptide nanotube wide band gap semiconductors (FF-PNTs). Diphenylalanine (FF), a short peptide consisting of two naturally occurring amino acid phenylalanine, can self-assemble into micro and nanosized tubular structures. The resulting organic self-assembled nanotubes are the FF-PNTs, which are stable/robust, rigid, and also biocompatible. They can be used in applications requiring the use of a wide bandgap semiconductor. FF-PNTs have been reported to have high thermal and chemical stability in addition to piezoelectric and pyroelectric properties. We show experimentally and theoretically that applying a longitudinal electric field allows the FF-PNT density of states to be tuned from a semiconductor to a metal, enabling effective charge transfer from the nanotube to the metal nanoparticles. This results in an enhancement in the state density of hot electrons. The effect is optimized through the physical alignment of the FF-PNTs, since the inherent electric dipoles of the FF-PNTs are then aligned and maximally responsive to

the applied longitudinal electric field (a cooperative effect). We demonstrate that this optoelectrical device enhances photocatalytic conversion for model oxidation reactions (exemplified here by p-aminothiophenol (PATP) oxidized to p-nitrothiophenol (PNTP), and 2-aminophenol (2-AMP) oxidized to 2-nitrophenol (2-NIP)), exploiting the facile field-activated trans-template charge transfer. We also demonstrate that this same approach can be used to enhance the strength of Raman scattering from low Raman cross-section molecules such as glucose and DNA-based molecules, establishing the potential of our template design for sensitive detection and analytics. This approach is versatile and can be applied to a range of plasmonic metal nanoparticle and semiconductor combinations.”

Overall, we have designed and implemented a novel nanostructured template/chip to facilitate charge transfer to target molecules, enhancing plasmonic photocatalytic activity and also enhancing Raman signal intensity, controlled by an applied electric field.

Finally, we comment on the relation to previous work from our own research group. We previously published in [4] the technical details of the alignment procedure in forming FF-PNTs with Ag nanoparticles (a necessarily prerequisite for the current work). In [5] we discussed how such a template is SERS active, and consider UV irradiation; we do not discuss catalysis. Our electro-optical approach in this work is distinct (here we use an applied electric field, not UV, to manipulate the electronic structure of the template to achieve the desired functionality). We observe photocatalysis, not previously discussed by us in [4,5]. Supplementary Figure S9 now compares photocatalysis using UV activation, to our new electro-optical approach. In addition to being a different and more controllable activation mechanism, we also note that the present field-activated technique exhibits vastly increased performance in catalysis. This new work also goes further to demonstrate enhanced SERS sensing of otherwise spectroscopically ‘dark’ target molecules.

2. The author suggests that the LSPR-excited hot electrons and electrons transferred from FF-PNTs to Ag NPs contribute to the O₂ activation step, leading to the formation of PNTP. This formation leads to the change in the SERS spectra. Please explain if this hypothesis also applies to the variation in Raman modes noted in Fig. 7. For example, some of these Raman peaks seem to be enhanced, or suppressed in the presence of electric field (circled in red). Explanations should be given to the readers if these different modes were also due to the some catalytic processes, or some other types of chemical transitions. Clarifications should be given to distinguish and justify the origin(s) of the changes in the Raman modes. Otherwise, this hypothesis for the formation PNTP may be difficult to convince the reader.

We thank the Reviewer for their careful analysis of our data and for highlighting an important point. Discussion of this issue is given below and also in the revised manuscript, which has strengthened the work. We have modified Fig. 7 of the paper and included an additional figure – Supplementary Figure 24 – specifically addressing this point. The observed changes (consistent with the literature) arise from an oxidation reaction catalysed by the template. An example of this, shown in Supplementary Figure 24 (b), is adenine which shows very strong peaks at 1050, 1661 and 1633 cm⁻¹ (circled), not present when the electric field is switched off. Literature studies (Phys. Chem. Chem. Phys. 2011, 13, 3851–3862 and J. Phys. Chem. C 2015, 119, 15, 8191-8198) report the same changes in Raman band positions when adenine is oxidised to oxoadenine. Similarly, oxidation of guanine

results in a new band at 684 cm^{-1} (see J. Phys. Chem. C 2015, 119, 15, 8191-8198). Similarly for cytosine.

Therefore, these results show that our template is not simply enhancing SERS sensing of these molecules, but also catalysing oxidation reactions. We have therefore removed these examples from Fig. 7 of the main paper, and included a discussion of the joint effect instead in Supplementary Figure 24. Note however that in general, all peak intensities are enhanced by the use of our template when a field is applied.

- 3. In the manuscript, the author stated that the hot electrons are transferred to adsorbed O_2 molecules, generating triplet diradical 3O_2 formed in degenerate antibonding π^*_{2px} and π^*_{2py} orbitals, that subsequently is involved in the oxidation of PATP to DMAB. Please comment on the field-assisted splitting of water molecules possibility due to the small gaps [6] formed from these FF-PNT/Ag NP, which could also assist the oxidation process. These small gaps in the FF-PNT/Ag NP could lead to intensively high electric field, even though the applied voltage could be low. In addition to the variation in O_2 concentration, since the system could be viewed as an integration of numerous nano/micro- scaled electrolysis cells, please also discuss on the variations of concentration in different components (that is PATP, PNTP, DMAD, 2-AMP, and 2-NIP) in terms of electroreduction as well.*

Our work is not electrochemical – experiments are performed in a dry environment, not solution. The system we study is not therefore to be regarded at an integration of electrolysis cells. Field-assisted splitting of water molecules is undoubtedly an interesting issue in the context of electrochemical systems, but it not important in the present work.

The referee notes that variations of concentration in different components in terms of electroreduction. Inspection of the spectra shows that with initial onset the electric field enhances the reagent SERS spectrum for both DMAP (Fig 2a) and 2-AMP (Fig 6a). However as the product Raman bands appear, the reagent Raman bands simultaneously drop. Therefore the oxidation reaction is more or less standard in the sense that oxidation product concentration increases proportionately to the decrease in reactant concentration, as evidenced by Raman band intensities.

- 4. Eight-fold increase in SERS intensity is observed with the FF-PNT/Ag NP template with electric field in comparison with that in the absence of an electric field. The author attributed this increase to the better charge transfer of the LSPR-excited hot electrons. Discussion should also be included about the aggregation and variation in the geometrical topologies and morphologies of the Ag nanoparticles and FF-PNT/Ag NP template under the influence of electric field, as the SERS measurements are performed on analyte solutions. These changes will affect the extinction coefficient (not just based on absorbance alone) which could lead to the difference/augmentation in the SERS enhancements.*

Firstly, we again emphasize that the experiments are performed in a dry environment, not in analyte solution. However, to address the substantive point raised by the Reviewer about the aggregation and variation in the geometrical topologies and morphologies of the Ag nanoparticles and template under the influence of the electric field, we now present an

extended discussion in Supplementary Figure S2. Importantly, we note that there is no change in the topology or morphology before and after application of the electric field.

The following has been added to the main paper to clarify this point. "SEM imaging of the sample showed spherical features assigned to Ag NPs. These Ag NPs occur in clusters predominately located above and between the FF-PNTs. We note that following the application of the probe molecule, these structural features remain unchanged. Following application of the electric field, recorded SEM images also showed no signs of changes in geometrical topologies and morphologies of the Ag NPs and FF-PNT/Ag NP template."

- 5. Figure S1 seems to be misleading to the reader. In Fig. S1c, the area of Si substrate deposited with FF-PNT/Ag NP shows aligned lines for locations without UV/ozone exposure. This gives the impression to the reader that FF-PNTs could be aligned on Si substrate even without exposure to UV/ozone. Maybe the author could consider demarcating the areas where the FF-PNTs are aligned/randomly orientated.*

We have improved Fig 1a and also Supplementary Figure S1 to make this issue clearer. The physical mask has an opening of 1mm, so the SiO₂ forms in this region. The gold electrodes have an opening of 0.1mm. The FF-PNTs are deposited only on the SiO₂ region (they are aligned on the SiO₂ as described in [4]). This is essential not just for the alignment, but also because the PNTs must be only on the insulating SiO₂ substrate rather than the semiconducting Si substrate.

Reply to Reviewer #3

We thank the Reviewer for their careful reading of the manuscript and their thorough analysis. The Reviewer notes that “Overall, it’s an interesting idea to select a bandgap matching semiconductor material and integrate it with the plasmonic reactor/sensor platform. Not like the conventional metal/TiO₂ or similar hybrid substrate, the described FF-PNT-AgNP substrate provides a new way of electrical control on the reaction or sensing”. Furthermore, the Reviewer states that “this work should be interesting to Nature Communications readership” – albeit after clarification of the points below. We have addressed each of these below and made suitable changes to the manuscript, as well as greatly expanding the Supplementary Information. We believe this has significantly strengthened the paper, and hope that publication is now possible.

1. *Electrochemical SERS (EC-SERS) and catalysis are very common technologies in these days, why should we use FF-PNT or semiconductor-plasmonic hybrid platform? We would directly combine TiO₂ like photon-active semiconductor with plasmonic materials to gain a simple use case with only light illumination to active catalysis and strong sensing, etc. The describe FF-PNT-AgNP platform actually needs both light illumination, a high AC voltage (10V) and a DC voltage. Perhaps it may have a better control, but its use is very complex. We usually develop a tailored technology to facilitate the use, but not complicate it. In an EC-SERS, a simple and small bias works for both sensing and catalysis. I don’t really get the point of authors’ claim on “our results point in the direction of future rational design of templates tailored for specific use”.*

First, we wish to highlight right away that our work is not electrochemical – all experiments are performed in a dry environment. We have now made this point clear in the revised manuscript. The electrochemical protocol described by the Reviewer is indeed the classic paradigm; but here we are presenting a new methodology.

Although the Reviewer suggests that our ‘tailored technology’ is complex to use, actually it is very simple in practice. UV illumination is not required for catalysis; the required electric field is simply generated by applying a voltage across electrodes (the electric field is then naturally and automatically aligned in the axial direction of the FF-PNTs in the template). No complex electronics are required. The benefit of our approach is greater control and efficiency/yield in catalysis relative to other methods, and enhanced sensing.

Our statement regarding rational design of functional template systems refers to optimizing nanocomposite systems for use in either catalysis or sensing, based on our understanding of the electronic properties of the components. For example, the field-activation and control of photocatalysis in our device results from the quasi-1d geometry, with the bandgap of the semiconductor substrate matched to the bandwidth of the nanoparticles.

We have revised the manuscript to reflect the above comments and address the point made by the Reviewer. We now emphasize more clearly the novel aspects of our work, how it is distinct from previous electrochemical studies, and which benefits are conferred by our template design. Specifically, we focus on nanocomposites combining plasmonic and

semiconductor materials in a dry environment. We highlight the use of the a carefully selected semiconductor and plasmonic metal combination: our theoretical simulations demonstrate the need for a quasi-1d semiconductor and for bandgap/bandwidth matching for optimized field-effect control. Experimentally, we show that peptide nanotubes possesses these required properties with respect to Ag nanoparticles. Above a threshold applied field, a facile route for charge transfer across the template is activated, enhancing SERS and plasmonic catalysis of target molecules. This is a novel process, distinct from methods such as electrochemical (EC)-SERS. These issues are now discussed in the paper's introduction:

“Nanocomposites combining semiconductor materials with metal NPs bring additional benefits to plasmonic photocatalysis. The contact potential difference between metal NPs and a semiconductor can separate photogenerated electrons and holes, thereby reducing electron-hole pair diffusion lengths and leading to more efficient photogenerated charge separation and transfer, which in turn enhances photocatalytic activity. Here we increase the probability and efficiency of both chemical reactions and SERS detection through electro-optical synergy, using a microfabricated chip design (in air rather than electrochemical). This is achieved through the use of a plasmonic-semiconductor system based on aligned diphenylalanine peptide nanotube wide band gap semiconductors (FF-PNTs). Diphenylalanine (FF), a short peptide consisting of two naturally occurring amino acid phenylalanine, can self-assemble into micro and nanosized tubular structures. The resulting organic self-assembled nanotubes are the FF-PNTs, which are stable/robust, rigid, and also biocompatible. They can be used in applications requiring the use of a wide bandgap semiconductor. FF-PNTs have been reported to have high thermal and chemical stability in addition to piezoelectric and pyroelectric properties. We show experimentally and theoretically that applying a longitudinal electric field allows the FF-PNT density of states to be tuned from a semiconductor to a metal, enabling effective charge transfer from the nanotube to the metal nanoparticles. This results in an enhancement in the state density of hot electrons. The effect is optimized through the physical alignment of the FF-PNTs, since the inherent electric dipoles of the FF-PNTs are then aligned and maximally responsive to the applied longitudinal electric field (a cooperative effect). We demonstrate that this optoelectrical device enhances photocatalytic conversion for model oxidation reactions (exemplified here by p-aminothiophenol (PATP) oxidized to p-nitrothiophenol (PNTP), and 2-aminophenol (2-AMP) oxidized to 2-nitrophenol (2-NIP)), exploiting the facile field-activated trans-template charge transfer. We also demonstrate that this same approach can be used to enhance the strength of Raman scattering from low Raman cross-section molecules such as glucose and DNA-based molecules, establishing the potential of our template design for sensitive detection and analytics. This approach is versatile and can be applied to a range of plasmonic metal nanoparticle and semiconductor combinations.”

We also supply further details regarding the photocatalytic activity of our template. We have included data in Supplementary Figure S9 comparing our electro-optical approach to a standard photocatalysis strategy based on super band gap irradiation of the semiconductor. We show that there is an advantage in better yield through the application of our electro-optical methodology, as well as excellent controllability, reproducibility and robustness.

2. *What's EL-SERS? What's the abbreviation of EL? Is it different to the electrochemical SERS?*

We used the abbreviation 'EL-SERS' to indicate SERS measurements in the presence of an electric field (in contrast to 'EC-SERS' for electrochemical SERS). But since this is non-standard terminology, we have removed this in the revised manuscript for clarity. Note that we are not using EC-SERS since experiments are performed in a dry environment.

3. *I find it's very strange to use the Si substrate, especially without a thick electrical passivation layer, e.g. SiO₂. Although I'm not very sure, it seems the two Au electrodes are shorted by the Si substrate. The resistance of the FF-PNT layer is similar to that of the Si substrate from the resistances shown by the authors. The SiO₂ passivation layer only appeared at the gap of electrodes and was formed by UV/Ozone. What's the thickness of the SiO₂? Does it thick enough to avoid the electric leakage?*

We have revised Figure 1 and Supplementary Figure S1 to clarify that the FF-PNTs are only deposited on a thick insulating SiO₂ passivation layer (which is certainly thick enough that there is no electrical leakage there). The SiO₂ layer forms in the gap ~1mm between the physical masks upon expose to UV/ozone. Electrical conductivity studies are shown in Supplementary Figure S3. The SiO₂ capped Si substrate is important for alignment of the FF-PNTs, which in turn is required for our quasi-1d template design so that the applied electric field is along the PNT axis. We have clarified this in the revised paper and Supplementary Information. See also answer to comment 6, below.

It is however correct that the gold electrodes are deposited on the underlying Si substrate. The electrodes are not shorted by the Si substrate (see Fig S3) but there will be some leakage. This means in practice that we have to use a greater voltage than otherwise required. Note however, that this does not account for the threshold voltage for catalytic activity, which is demonstrated as a genuine physical effect, not related to the substrate, from our theoretical calculations. A refined design for the template could have an SiO₂ passivating layer under the electrodes as well as under the PNTs to avoid such a complication.

4. *How AgNPs disperse on the surface with or without FF-PNT is not that clear? From the SEM pictures in the manuscript, they were very isolated from each other, meaning no current at biased AC/DC voltage. However, in Fig. S9b, the DC offset did increase the intensity of the PATP Raman. Why this can happen? Was there some extensive aggregations of AgNPs, which may form a kind of electrically conductive network?*

We have now updated the SEM image data in the paper, and included more detail on this in Supplementary Figure S2. This shows that the AgNPs disperse on the surface in a less ordered fashion without FF-PNTs. In general, the topology and morphology of the NPs on the PNTs comprise mixtures of dispersed and aggregated NPs. We have now included SEM images of more typical patternings. We found in all cases that a current flowed above the threshold bias. As the referee carefully notes, in Fig. S10b (which was previously Fig 9b before revisions), the DC offset increases the intensity of the SERS signal of PATP (note

however that this figure relates to control experiments for NPs on SiO₂ rather than the FF-PNTs). While signal enhancement was observed, there is far lower charge conductance on the SiO₂ substrate and so no catalytic activity. Conductance in that case could indeed, as the Reviewer suggests, be due to formation of an electrically conductive network of NPs (see S10b). By contrast, charge transfer is through the FF-PNTs when the applied field is above threshold.

5. This comes to my biggest question on this manuscript: were the measurements done in air after drying of the added sample droplet or directly in a sample solution? This important detail was not mentioned at all.

As now clearly stated all measurements were done in a dry environment, not electrochemical.

6. Another question related to the use of Si substrate. Si can absorb the excitation photons and sequentially forms carriers, which may have an impact on the electrochemistry at the Ag NPs. Have the authors been aware of this issue?

The band gap for SiO₂ is in the UV ($E_g > 8.5$ eV). Since the SiO₂ layer is the top layer, the Si substrate does not absorb the Raman excitation photons. The Raman spectra of the template with SiO₂/Si does not show the strong silicon Raman band, demonstrating that we have an optically thick SiO₂ top layer (photons do not reach the Si layer or scatter).

7. Why use an intense AC voltage together with a weak DC offset? From the result, the DC voltage is used for the oxidation. In the method section, the authors said the frequency has a small impact on the photochemical reaction. What's the role for the AC voltage? Will it still work by adding a high DC voltage like 4V? It should be explained somewhere in the content.

As shown and discussed in relation to Supplementary Figure S7, while DC bias alone is sufficient for catalysis of the oxidation reaction, the additional AC voltage is required for optimal performance (ie, the AC voltage significantly increases yield). To find the optimal conditions, we performed a large number of studies combining AC and DC electric field strengths (summarized in the Supplementary Information). In the main paper, we largely stick to using the optimal parameters. In particular, we found that a DC bias offset of c.a. >100 mV was required for good performance (as mentioned in the main paper), with the effect of applied offset maximised at c.a. 350 mV (Fig 6b). However, further increasing the offset causes no further enhancements, indicating saturation; around c.a. 30V the SERS spectral intensity was found to decrease again. So there is no advantage in increasing the offset above 350 mV (and high DC voltages have the disadvantage of heating effects).

8. In the frequency testing of AC voltage, such as Fig. S6a and Fig. S14a, the DC offset was 1 mV and the AC was 1V. I can understand this is a control experiment. But why not do the frequency testing at the typical working parameter, such as AC 6 V, DC 300mV? How does the frequency of AC voltage influence the reaction/biosensing with this setting? I think this is a more important control experiment, which should be added in the manuscript.

The DC offset at 1 mV in Fig. S6a and S14a (now S20) was a typo, which has now been corrected in the figures. It should have read 100 mv. We have now adapted Supplementary Figure S7 to show that at typical operating voltage and offset, a frequency of 40 KHz is optimal.

We keep the AC frequency at 40KHz for all measurements presented in the main paper.

9. The authors have ascendingly scanned AC voltages from 0 to 60 V. How about descending scan from 60 V back to 0 V? And how about the DC scan? It may indicate some charge trapping in the FF-PNT-AgNP-molecule system.

We tested this and found no observable variation: the process appears fully reversible. We undertook such experiments several dozen times and found no effect other than the reported relaxation. These data are shown in Supplementary Figure S8.

10. How many devices have been tested? I suppose every measurement of a new sample needs a new device/template to avoid any possible cross-talk contamination. Can the self-alignment technology of FF-PNT and the Ag NPs coating always give the same or similar resistance? What's the reproducibility and the uniformity of such processing? How about the device/template variation in the entire work? Is the minimum required 100mV DC offset always the same for every device/template in use?

This is indeed an important question about the robustness and reproducibility of our presented results. In fact, we produced and tested many different samples and found a high degree of reproducibility. Details of this are now shown in Supplementary Figure S23. Since our device is microfabricated (with feature sizes on relatively large length scales $\sim 0.1 - 1$ mm) rather than nanofabricated, the substrate design lends itself to easy reproducibility (this is part of the utility of the design). Specifically, we find variability of less than 4% across different samples (S23). Our self-assembly based method is highly reproducible and always give similar resistance. We studied this through SEM and optical microscopy to assess the physical alignment and through repeated number of substrates (over 100) used in electrical field studies with SERS. We found that a minimal offset at >100 mV was indeed always required. This is also in keeping with the theory, which predicts a threshold field is required.

11. I'm not well convinced by the explanation of the electric field induced charge transfer between NT-NP-molecule. Electric field is always there, no matter the presence of FF-PNP. But the experimental observations were highly related to the use of FF-PNT. Should it be related to the local electrostatic potential? The simulation was a 1D configuration, and the potential on the PNT-AgNP was fixed. In practice, the template/device was a 2D or even 3D FF-PNT-AgNP-molecule. The electrostatic potential should change from one electrode to the other, even in the presence of both AC and DC voltages. I wonder, have the authors studied different locations? Were the threshold AC ($\sim 4V$) or DC voltages ($\sim 100mV$) dependent on various locations within the electrode gap? Were the SERS signals the same from various locations? Indeed, I have been aware the authors mentioned: "Uniformity and reproducibility tests were performed on the sample at different spots, with only a 5% change observed." In the section of Enhancing SERS sensing. But since this work not only discusses SERS

enhancement, but also photocatalysis, AC/DC threshold, etc. Perhaps, the authors should give more details on this part.

As the Reviewer guessed, the electric field discussed here is an external applied field related to the applied voltage across the electrodes. The potential therefore does indeed vary down the FF-PNTs. We did study different locations: these data are now discussed in more detail in Supplementary Figure S23. We find very good reproducibility, independent of spot position. In fact, the experimental spot size is rather large (c.a. 40 microns) so we observe a good average response. As for the theoretical simulations, we modelled explicitly the varying potential down the length of the tubes (this is actually critical to describing faithfully the underlying physics). In Figure 5f of the main paper, we show the combined (net) result from NPs at different positions, simulating the self-averaging present experimentally. The important result of those calculations is that still one requires a threshold field strength for trans-template charge transfer. For reproducibility issues, see reply to point 10 above.

12. In the theory described in Fig. 3, electrons can come from Ag NP and FF-PNT. To match the HOMO-LUMO gap and the Fermi level of Ag NPs, it requires an excitation at 532 nm. The same laser was used for Raman, and it became difficult to define the contributions from plasmonic hot electrons or electrons propagating inside the FF-PNT at the bias. Will it be possible to use a 785 nm excitation to mismatch the energy levels, avoid the plasmonic related charge transfer and only leave the contribution of the PNT. And 785nm can still support SERS for monitoring the reaction/sensing?

We chose to fix the laser wavelength at 532 nm, and did not look at 785 nm for the purposes of detuning as suggested by the Reviewer. However, we did perform a raft of experiments to carefully disentangle experimentally the contributions from the various sources. The data from various control experiments are shown and discussed in Supplementary Figure S14. In particular, we studied the peptide nanotubes without the metal Ag NPs present (therefore guaranteeing no contribution to the Raman signal from the metal NPs). In this way, we looked at each constituent part separately. We find that the FF-PNTs do not significantly contribute to the signal on their own; this supports our claim of SERS enhancement due to the cooperative action of the template, and hence that the observed effect is routed in trans-template charge transfer (as also suggested by theory). We assign the increases in SERS for the probe molecule on FF-PNT only to an enhanced chemical mechanism through a charge transfer from the nanotube directly to the molecule.

13. Clearly, the sample molecule PATP or PAMP can be oxidized in the described device/template. How about the FF-PNT? It's made by peptides. Will it be irreversibly damaged by triplet oxygen? The same question for the AgNPs in use. Is there any alternative solid-state semiconductor can be used to replace the organic semiconductor in the describe concept?

We performed repeated experiments on FF-PNT (in absence or presence of the Ag NPs and probe molecules), over a number of cycles of applying electric field followed by relaxation. While the desired reported functionality remained robust and essentially unchanged, we saw no evidence of degrading the FF-PNTs or NPs comprising the template itself. Had there

been such a degrading, it would have shown up in changes to the SERS spectra. We carefully monitored for this, but saw no such effect.

The approach should however be versatile, and other nanoparticle and 1D semiconductor combinations could be used to achieve similar functionality. There is also versatility in respect to the alignment methodology which is based on wettability and can be applied to a range of nanotube materials. The methodology of applying an electric field is also versatile. Theoretical simulations show that the important characteristic ingredient is that the density of states bandwidth of the plasmonic metal should be on order of the bandgap of the semiconductor. For example, silver NPs could be combined with PVDF 1D nanotubes; or gold NPs could perhaps be used with AlN 1D nanorods. This is a topic for future research. We have made a comment on the versatility in the conclusion of the paper.

14. Fig. S11. The UV-vis absorption spectra. Why cannot see the LSPR resonance (~400nm) of Ag NPs in FF-PNT/AgNP spectra in both (a) and (b). BTW, the different orders of the major and the insert are annoying for reading.

We have modified Supplementary Figure S15 (as it is now) to address this issue. Now the LSPR resonance can be clearly seen. We now use a single figure with appropriate scaling.

15. I'm not very familiar with the modelling part. The Conductance of electrons in the NT-NP-molecule route sounds like the minimum physics to describe the charge transfer. I'll need to rely on other reviewers to evaluate the simplification for supporting a valid physical model.

The theoretical modelling here is based on a fully quantum mechanical description of charge transport, using tight-binding approaches to model the template, and Green's function methods to calculate conductance. The model is indeed a reduced effective model, containing the minimum necessary to capture the physics. This deliberate methodological philosophy was employed to demonstrate the key elements of the template required to confer the observed functionality (ie, field-effect threshold to activity through facile trans-template charge transfer). The justification and derivation of the effective model is provided in the paper and methods section. The solution of the model by more or less standard means is exact. In the end, the use of the effective model is validated by the qualitative agreement with experimental results.

16. A small comment: vibrational Stark effect can shift the Raman peak position. However, it usually requires a very high electric field (> 1 MV/m), which is not the case in the described work. It may have another explanation.

We agree and have removed reference to the Stark effect here.

17. *Some small typo mistakes:*

*P19, "DC offsets in the range of 5 to 60 V..." should be "AC offsets..."
What's the distance between the two electrodes: 0.1 or 1 mm. The main content says
1 mm, but the SM says 0.1 mm.*

Many thanks: typos now corrected. Regarding the opening between electrodes: the gap is 0.1mm, but the opening the physical mask where SiO₂ is formed and the FF-PNTs are deposited is 1mm. This has been corrected and made clearer in the text, and also in Fig 1 and Supplementary Figure S1.

REVIEWERS' COMMENTS:

Reviewer #1 (Remarks to the Author):

The authors addressed the issues that I raised in the first draft. The revised ms is suitable for publication in my opinion.

Reviewer #2 (Remarks to the Author):

For this manuscript, this reviewer is fine with the authors' replies and has no additional comments about the manuscript.